# Arecaceae Seeds Constitute a Healthy Source of Fatty Acids and Phenolic Compounds

**DOI:** 10.3390/plants12020226

**Published:** 2023-01-04

**Authors:** Miguel Ángel Rincón-Cervera, Abdallah Lahlou, Tarik Chileh-Chelh, Svetlana Lyashenko, Rosalía López-Ruiz, José Luis Guil-Guerrero

**Affiliations:** 1Food Technology Division, ceiA3, CIAMBITAL, University of Almería, 04120 Almería, Spain; 2Institute of Nutrition and Food Technology, University of Chile, Macul 7830490, Chile; 3Chemical-Physical Department, Analytical Chemistry of Pollutants, University of Almería, 04120 Almería, Spain

**Keywords:** Arecaceae, seeds, fatty acids, phenolic compounds, antiproliferative activity, HT-29 cells

## Abstract

Seeds of most Arecaceae species are an underutilized raw material that can constitute a source of nutritionally relevant compounds. In this work, seeds of 24 Arecaceae taxa were analyzed for fatty acids (FAs) by GC-FID, for phenolics by HPLC-DAD and LC-MS, and for their antitumor activity against the HT-29 colorectal cancer cell line by the MTT assay. Lauric, oleic, and linoleic acids were the prominent FAs. Cocoseae species contained total FAs at 28.0–68.3 g/100 g seeds, and in other species total FAs were from 1.2 (*Livistona saribus*) to 9.9 g/100 g (*Washingtonia robusta*). *Sabal domingensis*, *Chamaerops humilis*, and *Phoenix dactylifera* var. *Medjool* had unsaturated/saturated FA ratios of 1.65, 1.33–1.78, and 1.31, respectively, and contained 7.4, 5.5–6.3, and 6.4 g FAs/100 g seeds, respectively. Thus, they could be used as raw materials for healthy oilseed production. Phenolics ranged between 39 (*Livistona fulva*) and 246 mg/100 g (*Sabal palmetto*), and of these, caffeic acid, catechin, dactylifric acid, and rutin had the highest values. (-)-Epicatechin was identified in most seed extracts by LC-MS. Hydroalcoholic extracts from five species showed a dose-dependent inhibitory effect on HT-20 cells growth at 72 h (GI_50_ at 1533–1968 µg/mL). Overall, Arecaceae seeds could be considered as a cheap source of health-promoting compounds.

## 1. Introduction

Arecaceae is a family of perennial flowering plants commonly known as palms. Most Arecaceae species are widely distributed in regions characterized by humid and warm climates in South America, the Caribbean, the South Pacific, and Southern Asia. Some Arecaceae species are economically relevant as sources of a variety of goods, mainly oils. *Cocos nucifera* (from warm areas around the world), *Elaeis guineensis* (from tropical Africa), *Elaeis oleifera* (from the Amazon area), and to a lesser extent, *Acrocomia aculeata* (from South America), are the main Arecaceae species used for oil production at industrial scale [1].

According to the USDA report “Oilseeds: World Markets and Trade”, the production of palm oil in the world was 75.5 mill MT in 2021, with Indonesia and Malaysia being the main producers, and the production of palm kernel oil (PKO) was 8.7 mill MT in 2021. (https://ipad.fas.usda.gov/cropexplorer/cropview/commodityView.aspx?cropid=4243000, accessed on 6 June 2022).

PKO is obtained from the kernels of palm oil (*E. guineensis*) and it is the main commercial Arecaceae oil from kernels currently available [2]. However, other Arecaceae seeds are usually discarded as by-products during fruit processing, as happens with those of date palm [3]. PKO shows important differences regarding the fatty acid (FA) profiles compared with palm oil (PO) obtained from the fruit of *E. guineensis*; PO is rich in palmitic (PA, 16:0) and oleic (OA, 18:1*n*-9) acids (44% and 39% of total FAs, respectively), whereas PKO contains mainly lauric (LaA, 12:0), myristic (MA, 14:0), and OA (48, 16, and 15% of total FAs, respectively) [4].

The FAs of seeds from some Arecaceae species have been previously reported (see Appendix A), and considerable differences among them were found. For instance, OA was the main FA found in the seeds of *Phoenix* species such as *P. dactylifera* [5,6], as well as in seeds of palms belonging to other genera such as *Livistona chinensis*, *Chamaerops humilis*, and *Archontopheonix cunninghamiana* [7,8,9]. However, reported data showed that the seed oils from other Arecaceae species are rich in medium-chain FAs (MCFAs), i.e., caprylic acid (CyA, 8:0) in *Butia capitata* [10], capric acid (CA, 10:0) in *Syagrus romanzoffiana* [5], and LaA in *Dypsis lutescens* [11]. Other seeds contain mainly long-chain saturated FAs (LCSFAs) such as *Washingtonia filifera* (39.2% PA) [12]. On the other hand, LC-polyunsaturated FAs (LCPUFAs) such as linoleic acid (LA, 18:2*n*-6) were found in high percentages in the seed oils of *A. cunninghamiana* and *Trachycarpus fortunei* [5].

The consumption of OA-rich foods within a balanced diet may contribute to the regulation of fat metabolism, body weight, and energy expenditure, as well as improving blood lipid profiles, reducing the risk of cardiovascular diseases (CVDs), and preventing or treating insulin resistance and type-2 diabetes mellitus [13,14].

MCFAs have a role as a quick energy source due to their passive diffusion through the enterocytes and further transportation to the liver. Some evidence suggests that MCFAs have several metabolic roles involved in glucose and lipid metabolism, regulation of hormone secretion, and heat production [15]. MCFAs have been recognized as potential targets for treating metabolic and neurological diseases, and MCFA-rich diets may lead to an increase in fat oxidation and energy expenditure in healthy adults [16]. Despite these data, most seed oils from Arecaceae species remain unanalyzed; thus, to screen novel Arecaceae seeds as sources of nutritionally relevant FAs such as OA and MCFA is timely.

Although many studies have been published regarding the phenolic compound content of date fruits, information regarding the occurrence of phenolic compounds in seeds of Arecaceae species is scarce in the literature. Most studies have been focused on the valorization of by-products from the date palm industry. Reported values for total phenolic content (TPC) in date kernels are highly variable, about ~10–390 mg/100 g seeds [17,18,19,20], and it has been described that date kernels contain higher amounts of phenolic compounds than date fruits [21]. Caffeoyl hexoside, catechin, cinnamic acid, coumaric acid derivatives, epicatechin, 5-*O*-caffeoylshikimic acid isomers, hydrocaffeic acid, isoramnethin, and proanthocyanidin dimers, have been reported in date seeds [20,21]. However, as detailed in Appendix A, besides *C. nucifera*, *Phoenix* spp., *S. palmetto*, and *S. romanzoffiana*, data regarding the phenolic compound profiles of seeds from Arecaceae species are unavailable in the literature.

Most Arecaceae seeds are considered as fruit by-products whose valorization is desirable. Thus, a search for the bioactive compounds and biological activities of seeds from this family is timely. Despite a few previous works on the antitumor effects of different seed extracts of *P. dactylifera* on some cancer cell lines, there is a lack of information regarding the antitumor potential of most seeds of Arecaceae species. The aim of this work was to characterize the FA and phenolic compound profiles of seeds from 24 selected Arecaceae species, as well as the in vitro antitumor activity of the hydroalcoholic extracts of such seeds against the HT-29 human colorectal cancer cell line.

## 2. Results

### 2.1. Fatty Acids

The total FA amounts and FA profiles of seeds are detailed in Table 1. The lowest FA contents (≤3 g/100 g) were found in seeds of *L. saribus* (1.2), *A. cunninghamiana* (1.3), *T. fortunei* (1.8), *H. forsteriana* 5A (2.6), *C. macrocarpa* (2.7), and *D. lutescens* (3.0). Most samples had FA contents between 4.0 and 9.9 g FA/100 g of seeds. Species of Cocoseae were notable for their high FA content: *C. nucifera* (68.3), *S. romanzofianna* (28.1), and *B. capitata* (28.0 g FA/100 g of seeds). Within the same species, significant differences (*p* < 0.05) were found in seeds of *H. forsteriana* (samples 5A and 5B), and seeds of *P. dactylifera* (samples 13A and 13 B). However, no significant differences were found among the four samples of *C. humilis* (5.5–6.3 g FA/100 g).

CyA, CA, LaA, MA, PA, stearic acid (SA, 18:0), OA, and LA were found in Arecaceae seeds in different proportions (Table 1). CyA and CA were found mainly in seeds of *B. capitata*, *C. nucifera*, *Arenga engleri*, and *L. saribus.* The remaining seeds analyzed in this work showed low or undetectable amounts of both FAs.

LaA was found in the seeds of all analyzed species except in *Chamaedorea oblongata*. LaA was the main FA found in 7 out of 11 analyzed seeds of species within the Arecoideae subfamily, and in just 3 out of 18 samples of species belonging to the Coryphoideae subfamily (*A. engleri*, *Livistona fulva*, and *L. saribus*). Within Chamaedoreae, the two *Chamaedorea* species greatly diverged in their LaA percentages (8.5% in *C. microspadix* vs. undetectable levels in *C. oblongata*). LaA reached the highest proportion in seeds of *C. macrocarpa* (52.5), followed by *C. nucifera* (51.0), *D. lutescens* (44.3), and *H. forsteriana* 5A (43.9%). MA was found in all analyzed seeds and at proportions ≥ 8.0% in most cases, except for *P. canariensis* (7.0) and *C. humilis* (5.6–7.8%). The highest proportions of MA were found in the seeds of *D. lutescens* (29.7), *C. microspadix* (20.0), *C. nucifera* (16.6), and *C. oblongata* (16.3%). Regarding PA, it was especially abundant in the two Chamaedoreeae species, *C. oblongata* (31.2) and *C. microspadix* (19.2%). Within the remaining species, PA ranged between 5.4 (*B. capitata*) and 15.8% (*A. cunninghamiana*). SA was detected in most species, ranging between 1.2 (*C. macrocarpa*) and 6.2% (*C. oblongata*). OA was found in all analyzed species, and in Coryphoideae the values were higher than 30% in most analyzed samples, with a few exceptions (*L. saribus*, *A. engleri*, *L. fulva*, and *Phoenix reclinata*), whereas it reached values lower than 30% in most Arecoideae seeds. The lowest OA proportions were detected in *C. nucifera* (6.1) and *D. lutescens* (6.5%), whereas *P. dactylifera* var. *Medjool*, *Sabal domingensis*, *P. canariensis*, *S. romanzofianna*, *C. humilis* 19 C, and *S. palmetto* showed the highest percentages (44.7–47.3%). As for LA, it ranged between 1.1 (*C. nucifera*) and 32.9% (*A. cunninghamiana*), and it was the most abundant FA in the seeds of the latter species.

Considering the unsaturated FA/saturated FA (UFA/SFA) ratio, values were between 0.08 (*C. nucifera*) and 1.98 (*P. canariensis*). Seeds of Arecoideae species were characterized by UFA/SFA ratios lower than 1.0, with the highest values found in *S. romanzofianna* (0.95) and *C. microspadix* (0.92). In contrast, seeds of Coryphoideae species generally showed UFA/SFA ratios higher than 1.0, with the only exceptions being *A. engleri* (0.37), *L. saribus* (0.42), *L. fulva* (0.80), and *P. reclinata* (0.88).

### 2.2. Phenolic Compounds

Up to 23 phenolic compounds were identified and quantified by HPLC-DAD in the analyzed seeds (Table 2). Identification and quantification were carried out using analytical standards for 21 compounds. Dactylifric acid and eriodyctiol were quantified as syringic acid and quercetin equivalents, respectively. The LC-MS system was used to confirm the structure of all identified phenolics by the HPLC-DAD analysis and to characterize additional phenolic compounds for which analytical standards were not available in the current study. By the LC-MS system, which is described in Appendix A, compounds were identified according to the *m/z* of their molecular ion and a characteristic fragment ion (Appendix A). Besides phenolic compounds, three organic acids were identified and quantified as gallic acid equivalents: quinic acid (a cyclic polyol), chelidonic acid (a dicarboxylic acid), and *trans*-aconitic acid (a tricarboxylic acid).

The total phenolic compound content plus the three identified organic acids in Arecaceae seeds quantified by HPLC-DAD ranged from 44.0 (*C. oblongata*) to 264.3 mg/100 g dry weight (dw) (*S. palmetto*). Some species showed values <70 mg/100 g dw (*C. microspadix*, *B. capitata*, *C. nucifera*, both *C. humilis* samples, *L. fulva*, *L. saribus*, and *Washingtonia robusta*), and three had more than 150 mg/100 g dw (*A. cunninghamiana*, *S. minor*, and *L. chinensis*).

Concerning organic acids, all were below 20 mg/100 g, and quinic acid reached the highest value in *S. palmetto* (16.2) and *C. humilis* 19B (19.9 mg/100 g). Phenolic compounds detected at concentrations higher than 20 mg/100 g were caffeic acid in *A. cunninghamiana*, *C. macrocarpa*, *D. lutescens*, and *L. chinensis*; dactylifric acid in *P. dactylifera*, *P. reclinata*, *Sabaleae* species, and *L. chinensis*; the flavonoid catechin in *C. oblongata*, *P. dactylifera*, and *A. engleri*; and the flavonoid glycoside rutin in *A. cunninghamiana*, *H. belmoreana*, *H. forsteriana, S. minor*, and *S. palmetto*. Other phenolic compounds found occasionally at concentrations higher than 20 mg/100 g were vanillic acid in *P. dactylifera* var. *Deglet Nour*; salicylic acid in *S. palmetto*; *trans*-coumaric acid in *P. canariensis*, *P. dactylifera* var. *Medjool*, and *P. reclinata*; rosmarinic acid in *A. cunninghamiana*; and three flavonoids: eriodictyol in *H. forsteriana* 5A, luteolin in *S. minor* and *S. palmetto*, and kaempferol in *A. cunninghamiana*.

### 2.3. Antiproliferative Activity against HT-29 Cells

The antiproliferative activity against HT-29 cancer cells was measured as cell viability percentage compared to that of the cells used as negative controls without seed extracts, whose viability was 100%. After a preliminary screening, such activity was found for the hydroalcoholic extracts of five species (*A. cunninghamiana*, *H. belmoreana*, *S. bermudana*, *C. humilis*, and *T. fortunei*) at the assayed concentrations after 72 h exposure (Figure 1A). Results at 48 h were approximately 20% lower than those obtained at 72 h (data not shown). None of the five extracts reached the 50% of cell growth inhibition (GI_50_) at 1200 µg/mL, with the most active seed extract at that concentration being the one from *H. belmoreana* (59.5% cell viability). The extract of *C. humilis* 19D was the most active at 1600 µg/mL (47.6% cell viability), and all five seed extracts exceeded the GI_50_ at 2000 µg/mL, with the most active extract being that of *C. humilis* 19D (14.4% cell viability).

Figure 1B shows the concentrations of each seed extract and the four assayed phenolic standards to reach the GI_50_. Whereas extracts of *A. cunninghamiana*, *T. fortunei*, and *H. belmoreana* were close to 2000 µg/mL, those of *C. humilis* (sample 19D) and *S. bermudana* were ~1600 µg/mL. However, all extracts were found to be less active than pure phenolics, whose GI_50_ concentrations ranged between 40 and 80 µg/mL. Rosmarinic and ferulic acids (GI_50_ at 40 and 43 µg/mL) showed the highest antiproliferative activity among the four assayed phenolics.

## 3. Discussion

### 3.1. Fatty Acids Content

In this work, the FA profiles of the seeds of *H. belmoreana*, *C. oblongata*, and *L. fulva* were characterized for the first time. Overall, the results on the total FA content and FA profiles of all analyzed species agree with previous results (see Appendix A). Small differences in the FA profiles and total FA content could be related to several factors such as soil type, nutrient supply, plant chemotype, and environmental temperature, as previously reported [22]. Additionally, the development stage of seeds has been highlighted as a factor able to modify the FA profiles [7]. Seeds in an earlier development stage can contain a higher proportion of OA, which in turn is further converted into LA in plants by FA desaturases (FADs) [23], resulting in a higher proportion of LA than OA in mature seeds, as was detected in this work. The main differences in the FA profiles between the results of the present work and previous ones are limited to only a few cases (see Appendix A).

Within the Areceae tribe, *A. cunninghamiana*, *C. macrocarpa*, and *C. lutescens* were the best sources of LA, LaA, and MA, respectively, among all analyzed species (*p* < 0.05). The FA profile of *A. cunninghamiana* analyzed in the current work agreed with the values reported by [7] and disagreed with data reported by [9] for *A. cunninghamiana* seeds collected in Nigeria, where OA was the main FA (48.3% of total FAs), followed by LAs (23.5%). *H. belmoreana* and *H. forsteriana* 5A had similar FA profiles, with a high LaA content (40.4–43.9% of total FAs). Both shared the same origin (a botanical garden in Portugal), whereas the seeds of *H. forsteriana* 5B growing in a botanical garden ubicated in a colder area (Berlin, Germany) showed a higher unsaturated degree with significantly larger proportions of OA and LA and lower of LaA compared with the other two *Howea* samples. On the other hand, the FA profiles of *H. forsteriana* seeds found in the current study were similar to those already reported [5].

Seeds of the two Chamaedoreeae species analyzed in this work (*C. oblongata* and *C. microspadix*) typically showed high proportions of PA, OA, and LA, which were previously reported [7], and *C. oblongata* had the highest amount of SA among all analyzed species (*p* < 0.05). Noticeable proportions of MA were also reported for some species of the *Chamaedorea* genus, such as *C. microspadix* and *C. radicalis* [7], as was the case with the two *Chamaedorea* species analyzed in the current study.

The total FA content of Cocoseae species was significantly higher than those of the remaining species from other tribes (*p* < 0.05). Typically, members of the Cocoseae tribe are considered as sources of industrial oils, as currently happens with the kernels of *E. guineensis* (about 50% oil content). Seed oils from Cocoseae species contain high proportions of LaA, showing also significantly higher proportions of CyA and CA than the remaining species, which is in good agreement with previous reports [24]. In this work, *B. capitata* and *C. nucifera* were the best CyA- and CA-producers’ species (*p* < 0.05). As for the *S. romanzofianna* sampled in this work, it contained higher OA and LA, and lower LaA, percentages than those previously reported [5,25,26]. Such differences in the FA profiles could be related to the climates where samples were collected, given that the area of collection in this work is characterized by a dry and Mediterranean-type climate with lower average temperatures and rainfall levels than those of the South America regions where *S. romanzofianna* is native and where it was collected to be analyzed in previous works.

Regarding the tribe Caryoteae, results for *A. engleri* were close to those previously reported [5], with the most significant differences being the lower and higher proportions of MCFAs (CyA and CA) and LaA found in the *A. engleri* sampled in the current work.

As for Phoeniceae species, the seed FA profiles for *Phoenix* spp. have been largely documented, particularly those for *P. dactylifera* because of the huge amount of date seeds generated as by-products in the food industry, with MA, PA, and LA being their main FAs. OA was reported as the most abundant FA (usually > 47% of total FAs), followed by LaA (up to 22% of total FAs) [6,27]. Seeds of other *Phoenix* species showed similar values regarding FA profiles, such as *P. canariensis* (33.0% LaA, 32.9% OA, 11.5% MA, 10.6% LA), or *P. reclinata* (34.8% OA, 21.7% LaA, 19.9% LA, 10.6% MA) [5]. All the reports detailed in Appendix A are in good agreement with the results found in this work: OA and LaA were the main FAs found in seeds of *P. dactylifera*, whereas the results for *P. canariensis* were similar to those already reported [28].

Concerning Sabaleae species, OA was reported as the main FA in seeds (31-46% of total FAs), followed by LA (14–36%) and LaA (5–22%). Other FAs found, usually at proportions higher than 10% of total FAs, in the seeds of Sabaleae species were MA and PA [5,7]. Such results agree with those for the four *Sabal* species analyzed in this work, but the result for *S. palmetto* obtained here for OA (47.3%) was higher than those previously reported for this species (see Appendix A).

As for Trachycarpeae species, *L. fulva* and *L. saribus* seeds showed significantly higher and lower proportions of LaA and OA, respectively, than other samples analyzed in this work belonging to the same tribe (*p* < 0.05). To our knowledge, there are no previous reports on the FA profiles of *L. fulva* seeds. Concerning seeds of *L. chinensis*, their main FAs (LaA, LA, and OA) were found within the range reported in other studies dealing with *L. chinensis* seeds [5,7,29].

The proportion of the main FAs in *C. humilis* samples analyzed in the current study (LaA, PA, OA, and LA) were within the expected range according to reported FA profiles of seeds from this species. However, reported data showed higher variability for OA (29–44%) that that found in this work (40–46%) [5,7,8]. Finally, both *T. fortunei* and *W. robusta* contain mainly OA, LA, and LaA, which agrees with previous results [5,7].

Currently, *C. nucifera* and *E. guineensis* kernels are used worldwide to produce LaA-rich oils. This FA has interest from both technological and nutritional sides; LaA has been described as a potent and selective antimicrobial compound against pathogenic microorganisms, but with low activity against beneficial human gut microbiota, suggesting that such a FA might have a role as a modulator of intestinal health [30]. Significant antimicrobial, antifungal, and antivirus effects have been reported not only for LaA as a free FA but also for monolaurin; the LaA-based monoacylglycerol structure promotes the interaction with functional groups of the microorganism membranes, probably due to their surfactant properties [31]. LaA is also targeted to synthesize specific structured lipids with beneficial effects for human health, for instance to trigger healthy outcomes in human gut microbiota [32]. Currently, the most common source of LaA is coconut oil, which is widely used as food or a food ingredient. Therefore, screening for new potential sources of LaA is needed, especially if such materials are currently discarded or underutilized, as is the case with most Arecaceae seeds. Among the analyzed samples in this work, seeds having high LaA proportions (~35% of total FAs) and total FA content ≥5 g/100 g of seeds included *B. capitata*, *H. belmoreana*, *A. engleri*, and *L. fulva*. Such seeds could become interesting candidates from which to obtain LaA with the purpose of elaborating innovative products for human or animal healthcare.

Values for the UFA/SFA ratio are relevant when searching for novel oils and fats with specific technological or nutritional properties. An UFA/SFA ratio <1 in the analyzed seeds indicates high LaA, MA, and/or PA amounts, which have melting points of 43.2, 54.4, and 62.9 °C, respectively [33]. It means that lipids rich in such FAs would be solid at room temperature and, thus, have potential use as ingredients in shortenings and spreads. From a nutritional point of view, the dietary guidelines of the US Department of Agriculture recommend obtaining less than 10% of the daily energy intake from SFAs, for the general population, and the replacement of SFAs with UFAs when possible [34]. In this regard, a UFA/SFA ratio >1 was found in many of the analyzed seeds in the current work. Among them, seeds of *S. domingensis* (UFA/SFA = 1.65; 7.4 g FA/100 g seeds), *C. humilis* (UFA/SFA ratio 1.33–1.78; 5.5–6.3 g FA/100 g seeds), and *P. dactylifera* var. *Medjool* (UFA/SFA = 1.31; 6.4 g FA/100 g seeds), could become raw sources for healthy oils extraction.

### 3.2. Phenolic Compounds

The phenolic compound profiles of Arecaceae fruits have been previously reported. Conversely, little is known about the phenolic composition of Arecaceae seeds, and there are only some studies focusing on the seeds of *P. dactylifera*, *C. nucifera*, *S. palmetto*, and *S. romanzoffiana*. Most of these studies have reported values for TPC, which is usually measured by the Folin–Ciocalteau (F-C) method and expressed as mg of gallic acid equivalents per 100 g sample (mg GAE/100 g); however, data for the phenolic compound profiles of Arecaceae seeds are scarce, and reported compounds depend, among other factors, mainly on pure compound availability to be used as standards in HPLC analyses.

The reported phenolic compounds, when analyzing any plant tissues, greatly depend upon factors such as extraction method, developmental stage, growing conditions, soil type, geographical origin, plant varieties, and climate conditions (e.g., temperature, sunlight exposure, and water availability), and this fact is supported by previous evidence. For instance, for seeds of *P. dactylifera Goftar* and *Kabkab dalaki* varieties, the TPCs were 381 and 2838 mg GAE/100 g, respectively [35]. Furthermore, the effect of the developmental stage of seeds on TPC was assessed in seeds of *P. dactylifera* var. *Mozafati*. TPC values using the F-C method ranged between 1500 and 2500 mg GAE/100 g seeds; however, when HPLC was applied for phenolic compound quantification, much lower amounts of phenolics were detected: 15.14 and 55.36 mg/100 g seeds, depending on the developmental stage [19]. Such large differences in TPC values measured by the F-C method vs. those calculated as the sum of individual phenolic compounds by HPLC are due to the fact that the F-C methodology is rather unspecific, as the F-C reagent can also react with non-phenolic compounds such as sugars, ascorbic acid, and aromatic amines, among other compounds [36]. For this reason, more selective quantitative methods such as HPLC are preferred to explore the phenolic composition of any tissue. Furthermore, it was reported that the developmental stage of seeds influenced both the phenolic profiles and their total content, with gallic acid being the most abundant phenolic at earlier stages, whereas catechin was the predominant phenolic at later stages, and finally sinapic acid was the most abundant at the last ripening stages [19].

The effect of the solvent type (methanol, ethanol, water, or 80% acetone) on the phenolics extraction from seeds of *P. dactylifera* and *Serenoa repens* (saw palmetto) was assessed by [37]. They found that ethanol and acetone yielded the highest TPCs measured by the F-C method, with 1180 and 775 mg GAE/100 g for the ethanol and acetone extracts of *P. dactylifera* and *S. repens*, respectively. These authors identified nine phenolic compounds in the seed extracts of both Arecaceae species by HPLC-DAD, with protocatechuic acid being the most abundant phenolic at 110 and 142 mg/100 g for *P. dactylifera* and *S. repens*, respectively. Other phenolics were gallic, *p*-hydroxybenzoic, chlorogenic, caffeic, and syringic acids, as well catechin, while rutin and kaempferol were also found in *P. dactylifera* seeds. Total identified phenolics by HPLC-DAD were 300 and 164 mg/100 g for *P. dactylifera* and *S. repens*, respectively.

In the current study, the lack of knowledge about the phenolic profiles of Arecaceae seeds has been filled: 23 phenolic compounds and 3 organic acids were quantified by HPLC-DAD in seeds of 24 Arecaceae species belonging to 7 different tribes using analytical standards, while another 30 phenolic structures were identified by LC-ESI-MS/MS according to the *m/z* of their precursor ions and one of their characteristic fragment ions.

Concerning Cocoseae species, for *C. nucifera* the reported TPCs have ranged from 1.4 to 6–10 mg GAE/100 g fresh weight (fw) [38,39]. These studies found as main compounds caffeic, coumaric, gallic, hydroxy benzoic, and syringic acids, as well as catechin. In this work, the TPC for this species was higher than previous reports: 54.5 mg/100 g, and the main phenolics detected were catechin, rutin, quercetin, and kaempferol, although a large number of phenolic compounds were found at low amounts. Concerning *S. romanzoffiana*, several bis-stilbenes and stilbenoids have been reported [40] (Appendix A), while in this work 77.0 mg/100 g was computed as the sum of individual phenolics, of which the highest amounts were for salicylic and DL-*p*-OH-phenyllactic acids, and catechin.

Concerning Phoeniceae species reported here, the amounts of identified and quantified phenolic compounds for seeds of *P. dactylifera* varieties were similar (134.1 and 144.3 mg/100 g for *Deglet Nour* and *Medjool* varieties, respectively). Previous works on *Deglet Nour* showed disparate values for TPC, from 155 mg to 3 g GAE/100 g [27,41], and the latter indicated a total flavonoids content (TFC) at 2 g of rutin equivalents (RE)/100 g (Appendix A). Overall, our results agree with those by [27]. In *P. dactylifera* var. *Deglet Nour*, the most cited compounds have been caffeic, gallic, protocatechuic, and syringic acids, as well as flavonoids such as catechin and rutin [17,27,42,43]. All these compounds were found in the present work, but the main compound detected here was vanillic acid (22.0 mg/100 g). Reported phenolics for *P. dactylifera* var. *Medjool* were similar to the ones of the previous variety: caffeic, catechin, gallic, and syringic acids, as well as rutin [43], which were also found in this work among a wide variety of phenolics, along with high amounts of dactylifric and *trans*-coumaric acids (31.2 and 27.2 mg/100 g).

In this work, TPC of the seeds of *P. canariensis* and *P. reclinata* were lower than those of *P. dactylifera* and very close between them (88.4 and 89 mg/100 g). For *P. canariensis*, the values previously reported (91–403 mg/100 g) were slightly higher than those found in this work [44]. The main phenolics reported for this taxon were caffeic, coumaric, and gallic acids, as well as two flavonoids: rutin and naringenin [44,45]. In this work, for *P. canariensis*, among several minor compounds, caffeic and *trans*-coumaric acid amounts were the highest (16.2 and 28.1 mg/100 g, respectively), while for *P. reclinata*, dactylifric and *trans*-coumaric acids were the main phenolics (29.1 and 20.9 mg/100 g, respectively).

As for Sabaleae species, seeds of *S. palmetto* have been previously analyzed for phenolic compounds. One study reported total phenolic acids at 25–840 mg GAE/100 g, and TFCs at 11–207 mg as catechin equivalents (CE)/100 g dw [37], and the results of this work were within this range (246 mg/100 g). For this species, a previous work indicated the occurrence of phloretin glucosides, epicatechin, methyl gallate, and protocatechuic and *p*-*h*ydroxy-benzoic acids [46], while other authors reported caffeic, chlorogenic, gallic, *p*-*h*ydroxy-benzoic, protocatechuic, and syringic acids, as well as catechin [37]. In this work, three out of four species (*S. bermudana*, *S. minor*, and *S. palmetto*) showed significantly higher contents of phenolic compounds than all other analyzed species, and differences in the phenolic profiles were identified: dactylifric acid was found in all four Sabaleae species, and it was the most abundant phenolic in seeds of *S. bermudana* and *S. domingensis*. In fact, *S. bermudana* had the highest dactylifric acid concentration among all studied species, while rutin was the most abundant phenolic in *S. minor* and *S. palmetto*. Furthermore, the highest amount of salicylic acid among all analyzed species was found in *S. palmetto* (33.5 mg/100 g), and the highest amounts of luteolin were found in *S. palmetto* and *S. minor* (55.6 and 24.6 mg/100 g of seeds, respectively).

Other tribes lack previous reports about the phenolic-analyzed species. For the Trachycarpeae, *L. chinensis* showed the highest TPC (140.2 mg/100 g), with the most abundant being caffeic (66.1) and dactylifric acids (39.1 mg/100 g).

Among the Areceae, the highest amount of phenolic compounds was found in seeds of *A. cunninghamiana* (197.5 mg/100 g), and the most abundant phenolics were rosmarinic and caffeic acids (41.8 and 27.9 mg/100 g). Caffeic acid was also the most abundant phenolic in *D. lutescens* (53.5 mg/100 g).

Overall, four main phenolic compounds were detected in Arecaceae seeds: dactilyfric, caffeic, and rosmarinic acids, as well as rutin. Dactilyfric acid has been cited in date fruits [47]; however, this is the first report on the occurrence of dactylifric acid in Arecaceae seeds. Dactylifric acid (3-O-caffeoylshikimic acid) can be hydrolyzed into shikimic acid under certain conditions, and healthy properties have been reported for shikimic acid: neuroprotective, anti-inflammatory, anti-thrombogenic, and antibacterial cellular senescence prevention effects [48]. Caffeic acid is a hydroxycinnamic acid derivative which is receiving increasing interest as a bioactive compound for the management of several diseases such as metabolic syndrome, cancer, and diabetes, given its antidiabetic, anti-inflammatory, antioxidant, hypolipidemic, and hypotensive properties [49,50]. For rosmarinic acid, there are reports on its pharmacological effects, including antioxidant, antimicrobial, anti-inflammatory, antihyperglycemic, anticancer, hepatoprotective, cardioprotective, and neuroprotective ones [51,52,53]. As for the flavonoid glycoside rutin, it is considered an active phytochemical having cardioprotective, anti-inflammatory, anticancer, anti-allergic, antioxidant, and antidiabetic properties [54].

Regarding phenolic compounds identified by LC-MS (Table 3), (-)-epicatechin was the most widely distributed phenolic among all analyzed samples, being found in 21 out of 29 seed extracts. (-)-Epicatechin is a secondary metabolite in plants found in foods such as cocoa, tea, and grapes, among others, with known antioxidant, cardio and neuroprotective, anti-inflammatory, anti-diabetes, and anticancer properties, while it also acts as a muscle performance enhancer [55]. The largest variety of identified phenolics was found in seed extracts of Trachycarpeae: *L. fulva*, *L. saribus*, *T. fortunei*, and *W. robusta*. These findings make it worth conducting further research to attempt the quantification of other relevant phenolics in Arecaceae seeds.

### 3.3. Antiproliferative Activity of Seed Extracts against HT-29 Cells

Data on the antitumor effects of Arecaceae seeds are scarce. The cytotoxic effect of a terpenoids-, steroids-, amino acids-, and FAs-containing hexane extract of *P. dactylifera* seeds was in vitro assessed against cancer cell lines. The MTT assay was effected at 24 h, and the GI_50_ (the concentration that reduces total cell growth by 50%) was found to be 769, 962, and >1000 µg/mL for the MCF-7, HepG2, and A-549 cell lines (from liver, lung, and breast carcinoma, respectively) [56]. Additionally, the antiproliferative effect of a water:methanol (1:1 *v/v*) extract from *P. dactylifera* seeds was assayed on Caco-2, HepG2, and MDA cell lines (from colorectal, hepatic, and breast carcinoma, respectively) by the MTT assay at 24 and 48 h. The highest GI_50_ in all cases was achieved at 48 h, with values of 191, 257, and 219 µg/mL for HepG2, Caco-2, and MDA cells, respectively [57]. A recent work reported that for a 95% ethanol extract from *P. dactylifera* seeds, the GI_50_ was 86, 100, and 122 µg/mL for the MDA-MB-231, MCF-7, and HepG2 cell lines, respectively [58].

In this work, only five hydroalcoholic extracts of the assayed taxa were active against HT-29 colorectal cancer cells (Figure 1), but at higher concentrations than those reported in previous studies. A dose-dependent response was observed in the five active extracts between 1200 and 2000 µg/mL (Figure 1A). No clear relationship could be established between the phenolic composition of seed extracts and the inhibitory effect on cancer cell growth according to the observed results, which could be explained by the activity of the hydroalcoholic extracts being due to a synergistic action among several compounds present in such extracts. All seed extracts had a GI_50_ against HT-29 cells much higher than that of the pure phenolics (3,4-dihydroxyhydrocinnamic, gallic, rosmarinic, and ferulic acids) (Figure 1B). Moreover, such values were also higher than those of the hydroalcoholic extracts of Arecaceae fruits, which reached GI_50_ between 135 and 200 µg/mL [47].

## 4. Materials and Methods

### 4.1. Samples and Chemicals

Information regarding the identification, location, and date of collection of the Arecaceae species whose seeds were sampled in this study is shown in Table 4. Upon reception in the laboratory, all samples were labeled, dried at 60 °C in an air-forced oven until constant weight, and kept in sealed plastic bags at −20 °C until analysis. Unless otherwise stated, all solvents and reagents were purchased from Merck (Madrid, Spain).

### 4.2. FA Analysis

The FA profiles were obtained by gas chromatography coupled with a flame ionization detector (GC-FID) after direct derivatization of the oil contained in seeds to FA methyl esters (FAME), as described in Appendix A [59]. Peaks were identified by retention times obtained for known FAME standards: 37-component FAME Mix (CRM47885) and methyl stearidonate ≥97% (43959) from Supelco (Merck, Darmstadt, Germany).

### 4.3. Identification and Quantification of Phenolic Compounds

Seeds were ground and extracted with methanol:water (60:40 *v/v*), and extracts were analyzed by high-performance liquid chromatography coupled with a diode array detector (HPLC-DAD). Phenolic compounds as well as three organic acids (quinic, chelidonic, and *trans*-aconitic acids) were identified according to their characteristic retention times and quantified by external calibration curves using analytical standards for each compound, except for organic acids, dactylifric acid, and eriodyctiol, which were quantified using gallic acid, syringic acid, and quercetin, respectively. A 280 nm-HPLC-DAD chromatogram of *Sabal minor* is depicted in Figure 2. Other phenolic compounds whose analytical standards were not available were identified by liquid chromatography coupled with a mass spectrometer (LC-MS) according to the *m/z* of their molecular ion and one of their characteristic fragments. The methodologies for extraction and analysis of phenolics are detailed in the Appendix A.

### 4.4. Antiproliferative Assays of Phenolic Extracts on the HT-29 Cell Line

The antiproliferative activity of the hydroalcoholic (methanol:water, 60:40, *v/v*) extracts of Arecaceae seeds was assayed against the HT-29 human colorectal cancer cell line as described by [60]. A detailed description of this methodology is given in the Appendix A.

### 4.5. Statistical Analysis

All samples were analyzed in triplicate. Data were assessed for normality using a Shapiro–Wilk test and submitted to one-way ANOVA, and the comparison of means was made using Duncan’s Multiple Range Test. Statistical analyses were performed using Statgraphics^©^ centurion XVI (StatPoint Technologies, Warrenton, VA, USA).

## 5. Conclusions

In this work, the seeds of 24 Arecaceae species were studied for FAs, phenolics, and antitumor activity. This is the first report on the FA profiles of *H. belmoreana*, *C. oblongata*, and *L. fulva*, and this is the first time most of these species have been analyzed for their phenolic composition and antitumor activity. The seeds of Cocoseae species analyzed in this work stand out for their high FA amounts, and they contain high proportions of LaA, showing also significantly higher proportions of CyA and CA than the remaining species. Among analyzed seeds, those from *S. domingensis*, *C. humilis*, and *P. dactylifera* var. *Medjool*, could become raw sources for oil extraction with interesting nutritional potential. Four main phenolic compounds were quantified in Arecaceae seeds by HPLC-DAD: dactilyfric, caffeic, and rosmarinic acids, as well as rutin. Among the compounds identified by LC-MS, (-)-epicatechin was the most widely distributed phenolic among the analyzed samples, being found in 21 out of 29 analyzed seed extracts. Such compounds are considered human-health promoters. The phenolics-containing hydroalcoholic extracts of five species showed dose- and time-dependent inhibition against the human colorectal cancer cell line HT-29. A dose-dependent response was observed in five extracts between 1200 and 2000 µg/mL. Overall, considering both the richness in bioactive compounds and antitumor actions, some of the seeds analyzed here, and especially the nutritionally relevant ones, could be used to extract valuable ingredients for use by the food industry to improve the nutritional value of several marketed foods. Other actions to be developed should focus on revealing changes in the concentrations of bioactive compounds and/or bioactivity in Arecaceae seeds over multi-year periods.

## Figures and Tables

**Figure 1 plants-12-00226-f001:**
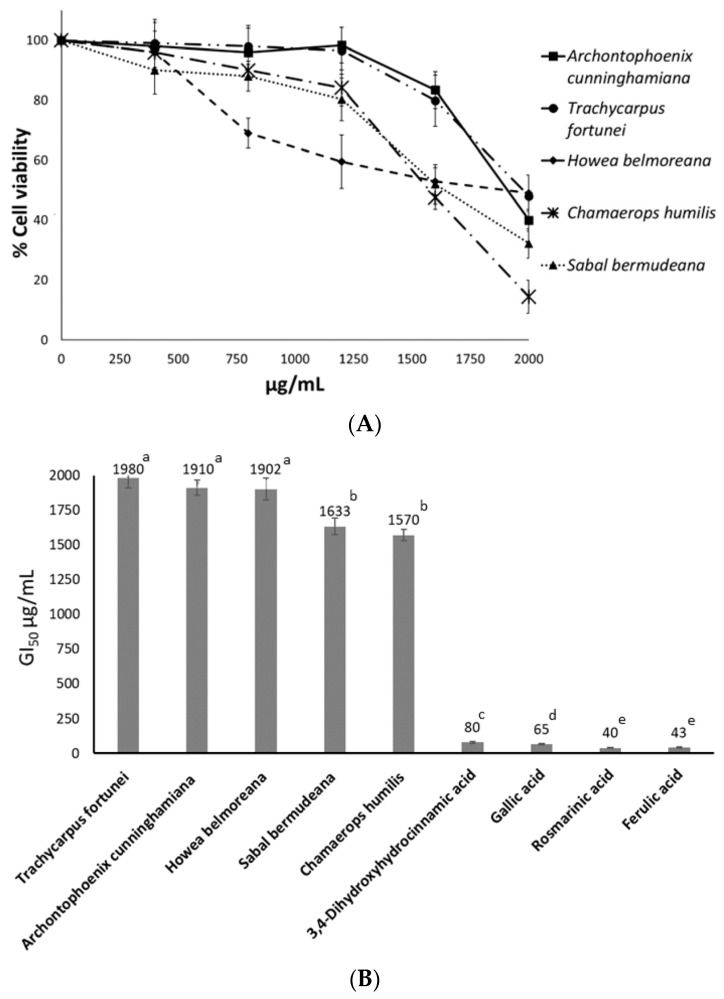
MTT assay. (**A**): Concentration-response plot for HT-29 cells after exposure to five Arecaceae seed extracts for 72 h. (**B**): GI_50_ after HT-29 cell exposure for 72 h to five seed extracts and to 3,4-dihydroxyhydrocinnamic, gallic, rosmarinic, and ferulic acids. Data represent the mean of three complete independent experiments ± SD (error bars). Above the bars, means followed by different letters are significantly different at *p* < 0.05.

**Figure 2 plants-12-00226-f002:**
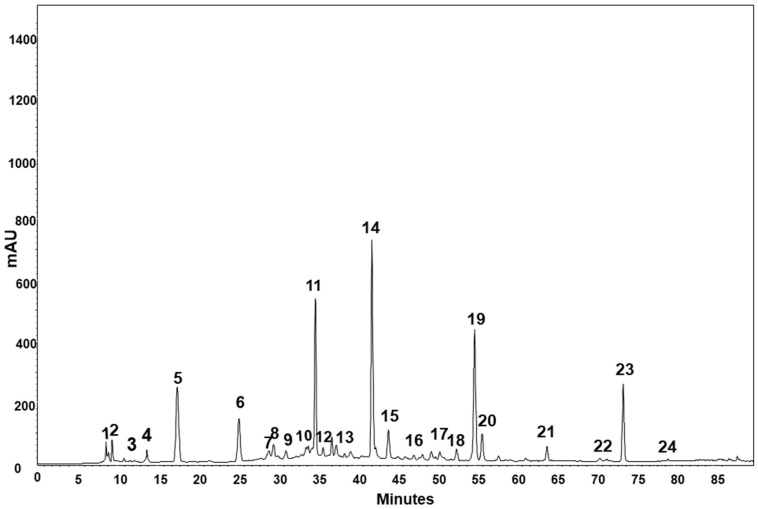
280 nm-HPLC chromatogram of the phenolic-containing water:methanol extract of *Sabal minor* seeds. 1. Quinic acid; 2. chelidonic acid; 3. gallic acid; 4. vanillic acid; 5. protocatechuic acid; 6. salicylic acid; 7. 4-hydroxybenzoic acid; 8. DL-p-Hydroxyphenyl lactic acid; 9. 3,4-Dihydroxyhydrocinnamic acid; 10. chlorogenic acid; 11. caffeic acid; 12. catechin; 13. syringic acid; 14. dactylifric acid; 15. *trans*-p-coumaric acid; 16. ferulic acid; 17. sinapic acid; 18. eriodictyol; 19. rutin; 20. rosmarinic acid; 21. 2-Hydroxy-4-methoxy benzoic acid; 22. quercetin; 23. luteolin; 24. kaempferol.

**Table 1 plants-12-00226-t001:** Fatty acid profiles of seeds from Arecaceae species ^a,b,c^.

Nr.	Species	FAs (% of Total FAs)								Total FA g/100 g Dry Seeds
		8:0 (CyA)	10:0 (CA)	12:0 (LA)	14:0 (MA)	16:0 (PA)	18:0 (SA)	18:1*n*-9 (OA)	18:2*n*-6 (LA)	
		Subfamily Arecoideae								
		Tribe Areceae								
1	*Archontophoenix cunninghamiana*	0.4 ± 0.0 ^h,i^	0.5 ± 0.4 ^g,h^	21.4 ± 0.5 ^j,k,l,m^	12.8 ± 1.8 ^d,e,f^	15.8 ± 2.5 ^c^	2.2 ± 0.1 ^h,I,j,k^	14.0 ± 2.2 ^m^	32.9 ± 2.2 ^a^	1.3 ± 0.3 ^m^
2	*Chambeyronia macrocarpa*	1.5 ± 0.0 ^e,f,g^	1.1 ± 0.4 ^e,f^	52.5 ± 0.4 ^a^	10.8 ± 1.1 ^g,h,i^	9.1 ± 0.5 ^h,I,j,k^	1.3 ± 0.4 ^m^	15.1 ± 1.7 ^l,m^	8.7 ± 0.7 ^p,q^	2.7 ± 1.4 ^j,k,l,m^
3	*Chrysalidocarpus lutescens*	n.d	0.8 ± 0.1 ^g,h^	44.3 ± 1.8 ^c^	29.7 ± 0.4 ^a^	9.7 ± 0.0 ^f,g,h,i,j^	1.7 ± 0.1 ^k,l,m^	6.5 ± 0.0 ^n^	7.4 ± 0.5 ^q^	3.0 ± 0.5 ^j,k,l,m^
4	*Howea belmoreana*	1.4 ± 0.1 ^e,f,g,h^	1.7 ± 0.2 ^e^	40.4 ± 1.2 ^d,e^	13.2 ± 1.7 ^d,e,f^	7.4 ± 0.1 ^l,m,n^	2.7 ± 0.4 ^f,g,h,i^	21.4 ± 1.7 ^k^	10.7 ± 1.1 ^l,m,n^	6.3 ± 0.9 ^d,e,f,g^
5A	*H. forsteriana*	1.4 ± 0.0 ^e,f,g,h^	2.3 ± 0.1 ^d,e^	43.9 ± 0.4 ^c,d^	13.9 ± 0.1 ^d^	6.2 ± 0.0 ^n,o^	2.6 ± 0.0 ^g,h,i^	21.2 ± 0.1 ^k^	7.8 ± 0.4 ^p,q^	2.6 ± 1.2 ^k,l,m^
5B	*H. forsteriana*	n.d	n.d	30.5 ± 0.4 ^h^	12.2 ± 0.1 ^e,f,g^	8.6 ± 0.5 ^i,j,k,l^	4.2 ± 0.1 ^b^	30.3 ± 0.2 ^hi^	14.1 ± 0.5 ^i,j^	6.7 ± 0.4 ^d,e,f^
		Tribe Chamaedoreeae								
6	*Chamaedorea microspadix*	0.5 ± 0.0 ^h,i^	0.6 ± 0.0 ^g,h^	8.5 ± 0.2 ^r^	20.0 ± 0.3 ^b^	19.2 ± 0.4 ^b^	2.6 ± 0.1 ^g,hi^	26.8 ± 0.2 ^ij^	20.6 ± 0.2 ^e,f^	4.8 ± 0.1 ^f,g,h,i,j^
7	*C* *. oblongata*	n.d	n.d	n.d	16.3 ± 0.1 ^c^	31.2 ± 0.7 ^a^	6.2 ± 0.5 ^a^	20.3 ± 0.4 ^k^	26.3 ± 0.1 ^c^	4.3 ± 0.7 ^g,h,i,j,k^
		Tribe Cocoseae								
8	*Butia capitata*	14.4 ± 1.3 ^a^	10.5 ± 0.1 ^a^	34.9 ± 1.5 ^f,g^	8.7 ± 0.1 ^j,k^	5.4 ± 0.1 ^o^	2.3 ± 0.3 ^g,h,i,j^	18.8 ± 0.6 ^k,l^	4.4 ± 0.0 ^r^	28.0 ± 0.9 ^b^
9	*Cocos nucifera*	7.4 ± 0.7 ^b^	6.8 ± 0.4 ^b^	51.0 ± 0.3 ^b^	16.6 ± 0.6 ^c^	7.9 ± 0.3 ^k,l,m^	2.7 ± 0.5 ^f,g,h^	6.1 ± 0.0 ^n^	1.1 ± 0.0 ^s^	68.3 ± 2.9 ^a^
10	*Syagrus romanzofianna*	3.5 ± 2.4 ^d^	2.6 ± 1.3 ^d^	25.2 ± 8.0 ^i^	9.0 ± 0.3 ^j,k^	9.8 ± 1.0 ^f,g,h,i^	1.3 ± 0.0 ^m^	39.5 ± 2.1 ^e,f,g^	9.3 ± 1.8 ^n,o,p^	28.1 ± 1.3 ^b^
		Subfamily Coryphoideae								
		Tribe Caryoteae								
11	*Arenga engleri*	5.6 ± 0.4 ^c^	4.9 ± 0.0 ^c^	37.4 ± 0.3 ^e,f^	9.4 ± 0.3 ^i,j^	12.3 ± 0.3 ^d,e^	3.3 ± 0.4 ^c,d,e^	15.6 ± 0.0 ^l,m^	11.2 ± 0.2 ^l,m^	8.4 ± 0.3 ^c,d^
		Tribe Phoeniceae								
12	*Phoenix canariensis*	n.d	n.d	13.4 ± 1.2 ^q^	7.0 ± 1.1 ^l,m^	13.0 ± 0.8 ^d^	n.d	45.5 ± 0.2 ^a,b,c^	20.5 ± 1.9 ^e,f^	3.7 ± 3.6 ^h,i,j,k,l^
13A	*P. dactylifera* var. *Deglet Nour*	0.4 ± 0.0 ^h,i^	0.5 ± 0.0 ^g,h^	22.0 ± 0.4 ^i,j,k,l^	11.1 ± 1.6 ^g,h^	9.7 ± 1.3 ^f,g,h,i,j^	3.2 ± 0.2 ^d,e,f^	42.5 ± 1.1 ^b,c,d,e,f^	8.9 ± 0.6 ^o,p,q^	3.4 ± 0.5 ^i,j,k,l,m^
13B	*P. dactylifera* var. *Medjool*	0.4 ± 0.0 ^h,i^	0.4 ± 0.1 ^g,h^	17.5 ± 0.1 ^n,o,p^	10.9 ± 0.0 ^g,h,i^	10.5 ± 0.0 ^f,g^	2.4 ± 0.6 ^g,h,i,j^	44.7 ± 0.2 ^a,b,c,d^	10.5 ± 0.0 ^m,n,o^	6.4 ± 1.8 ^d,e,f,g^
14	*P. reclinata*	0.3 ± 0.0 ^h,i^	0.4 ± 0.0 ^g,h^	23.2 ± 0.2 ^i,j,k^	14.2 ± 0.2 ^d^	12.8 ± 0.0 ^d^	2.2 ± 0.0 ^g,h,i,j,k^	27.5 ± 0.1 ^h,i,j^	19.0 ± 0.1 ^f,g^	3.8 ± 0.1 ^h,i,j,kl^
		Tribe Sabaleae								
15	*Sabal bermudana*	0.5 ± 0.0 ^h,i^	0.6 ± 0.1 ^g,h^	20.4 ± 0.2 ^k,l,m,n^	11.6 ± 0.0 ^f,g^	9.5 ± 0.0 ^g,h,i,j^	2.3 ± 0.0 ^g,h,ij^	41.5 ± 0.1 ^c,d,e,f^	12.1 ± 0.1 ^k,l^	4.3 ± 0.5 ^g,h,i,j,k^
16	*S. minor*	0.5 ± 0.1 ^h,i^	0.5 ± 0.0 ^g,h^	23.9 ± 0.1 ^i,j,k^	10.8 ± 0.0 ^g,h,i^	6.6 ± 0.3 ^m,n,o^	2.1 ± 0.3 ^i,j,k,l^	38.4 ± 0.4 ^f,g^	13.9 ± 0.1^j^	4.3 ± 0.4 ^g,h,i,j,k^
17	*S. palmetto*	n.d	n.d	15.9 ± 0.2 ^o,p,q^	11.0 ± 0.1 ^g,h,i^	9.7 ± 0.4 ^f,g,h,i,j^	2.3 ± 0.1 ^g,h,i,j^	47.3 ± 0.5 ^a^	13.9 ± 0.1^j^	5.4 ± 0.1 ^e,f,g,hi^
18	*S. domingensis*	0.2 ± 0.0 ^i^	0.2 ± 0.0 ^g,h^	14.7 ± 0.2 ^p,q^	10.0 ± 0.1 ^h,i,j^	9.8 ± 0.0 ^f,g,h,i^	2.4 ± 0.0 ^g,h,i,j^	45.8 ± 0.0 ^a,b^	15.7 ± 0.1 ^h,i^	7.4 ± 0.4 ^d,e^
		Tribe Trachycarpeae								
19A	*Chamaerops humilis*	n.d	n.d	18.2 ± 0.1 ^m,n,o,p^	7.8 ± 0.7 ^k,l^	9.8 ± 0 ^f,g,h,i^	3.8 ± 0.2 ^b,c,d^	41.1 ± 0.1 ^d,e,f^	18.4 ± 0.4 ^g^	6.3 ± 0.7 ^d,e,f,g^
19B	*C. humilis*	n.d	n.d	13.4 ± 0.1 ^q^	7.5 ± 1.5 ^k,l^	12.8 ± 0.5 ^d^	3.9 ± 0.2 ^b,c^	39.5 ± 0.1 ^e,f,g^	24.1 ± 0.4 ^d^	5.5 ± 0.3 ^e,f,g,hi^
19C	*C. humilis*	n.d	n.d	17.2 ± 0.2 ^n,o,p^	5.6 ± 0.0 ^m^	10.4 ± 0.1 ^f,g,h^	2.8 ± 0.1 ^e,f,g^	46.5 ± 0.5 ^a,b^	17.7 ± 0.1 ^g^	5.7 ± 0.1 ^e,f,g,h^
19D	*C. humilis*	1.9 ± 0.0 ^e,f^	2.7 ± 0.0 ^d^	21.2 ± 0.2 ^j,k,l,m^	6.7 ± 0.0 ^l,m^	8.4 ± 0.1 ^j,k,l^	1.9 ± 0.1 ^j,k,l^	42.9 ± 0.8 ^b,c,d,e^	14.2 ± 0.3 ^h,i,j^	6.1 ± 0.3 ^e,f,g^
20	*Livistona chinensis*	n.d	n.d	18.9 ± 0.3 ^l,m,n,o^	10.7 ± 0.4 ^g,h,i^	8.6 ± 0.4 ^i,j,k,l^	1.6 ± 0.1 ^l,m^	39.0 ± 0.7 ^e,f,g^	21.5 ± 0.1 ^e^	4.4 ± 0.3 ^g,h,i,j,k^
21	*L. fulva*	0.6 ± 0.0 ^h,i^	0.9 ± 0.0 ^f,g^	32.7 ± 1.5 ^g,h^	8.7 ± 0.4 ^j,k^	9.1 ± 0.4 ^h,i,j,k^	n.d	26.0 ± 0.6 ^j^	15.7 ± 0.9 ^h,i^	8.4 ± 0.3 ^c,d^
22	*L. saribus*	2.4 ± 0.3 ^d,e^	5.3 ± 1.3 ^c^	30.1 ± 1.4 ^h^	13.4 ± 0.6 ^d,e^	12.9 ± 0.5 ^d^	3.3 ± 0.4 ^c,d,e^	12.7 ± 0.6 ^m^	15.7 ± 0.9 ^h^	1.2 ± 0.1 ^m^
23	*Trachycarpus fortunei*	0.8 ± 0.0 ^g,h,i^	0.9 ± 0.0 ^f,g^	12.9 ± 1.1 ^q^	8.8 ± 1.0 ^j,k^	10.9 ± 0.2 ^e,f^	2.4 ± 0.3 ^g,h,i,j^	31.3 ± 0.8 ^h^	29.0 ± 0.9 ^b^	1.8 ± 0.1 ^l,m^
24	*Washingtonia robusta*	0.9 ± 0.0 ^f,g,h,i^	0.8 ± 0.0 ^g^	24.4 ± 0.2 ^i,j^	12.8 ± 0.0 ^d,e,f^	7.7 ± 0.0 ^l,m^	2.8 ± 0.0 ^e,f,g^	36.7 ± 0.8 ^g^	13.4 ± 0.2 ^j,k^	9.9 ± 0.3 ^c^

^a^ Results are shown as mean value ± standard deviation (n = 3); ^b^ Data sharing the same superscript letter/s in each column are not significantly different (*p* < 0.05) according to one-way ANOVA followed by Duncan’s test; ^c^ n.d.: not detected.

**Table 2 plants-12-00226-t002:** Phenolic compound and organic acid (quinic, chelidonic, and *trans*-aconitic acids) profiles (mg/100 g dry weight) for seeds from Arecaceae species ^a^.

Code	Species	Quinic Acid ^b^	Chelidonic Acid ^b^	*Trans*-Aconitic Acid ^b^	Gallic Acid	Vanillic Acid	Protocatechuic Acid	Salicylic Acid	4-OH-Benzoic Acid	DL-*p*-OH-Phenyllactic Acid
	*Retention time (min)*	*8.73*	*9.38*	*11.25*	*13.84*	*14.57*	*18.57*	*25.27*	*27.55*	*29.82*
		Subfamily Arecoideae								
		Tribe Areceae								
1	*Archontophoenix cunninghamiana*	2.3 ± 0.1 ^i,j,k^	1.1 ± 0.1 ^e,f,g^	n.d	4.8 ± 0.8 ^a^	0.2 ± 0.0 ^l,m^	9.8 ± 0.8 ^b,c^	11.9 ± 1.8 ^b,c^	4.2 ± 0.6 ^e^	1.8 ± 0.2 ^g,h^
2	*Chambeyronia macrocarpa*	4.3 ± 0.3 ^f,g,h^	1.2 ± 0.1 ^e,f^	0.6 ± 0.0 ^c,d^	0.2 ± 0.0 ^l^	2.7 ± 0.6 ^e,f,g^	0.8 ± 0.0 ^h^	2.9 ± 0.2 ^i,j,k,l^	2.6 ± 0.5 ^f,g,h^	1.2 ± 0.1 ^g,h,i^
3	*Dypsis lutescens*	1.5 ± 0.1 ^i,j,k,l^	0.2 ± 0.0 ^i,j^	0.7 ± 0.1 ^c,d^	0.1 ± 0.0 ^l^	5.2 ± 0.1 ^d^	7.8 ± 0.7 ^d,e^	10.0 ± 0.8 ^c,d,e^	9.2 ± 1.2 ^a^	15.8 ± 1.2 ^b^
4	*Howea belmoreana*	0.1 ± 0.0 ^l^	0.3 ± 0.0 ^h,i,j^	1.1 ± 0.1 ^c,d^	0.1 ± 0.0 ^l^	1.2 ± 0.1 ^h,i,j,k,l,m^	5.6 ± 0.7 ^f,g^	6.4 ± 0.2 ^f,g,h^	6.9 ± 0.7 ^c,d^	15.4 ± 0.2 ^b^
5A	*H. forsteriana*	0.1 ± 0.0 ^l^	0.8 ± 0.2 ^e,f,g,h,i,g,^	n.d	0.9 ± 0.1 ^f,g,h,i,j,k^	1.3 ± 0.1 ^h,i,j,k,l,m^	1.5 ± 0.2 ^h^	4.5 ± 0.1 ^g,h,i^	0.9 ± 0.2 ^l,m^	6.1 ± 0.2 ^f^
5B	*H. forsteriana*	0.1 ± 0.0 ^l^	0.1 ± 0.0 ^j^	n.d	2.9 ± 0.2 ^b^	1.8 ± 0.1 ^f,g,h,i,j,k^	0.9 ± 0.2 ^h^	3.8 ± 0.3 ^h,i,j,k^	3.9 ± 0.2 ^e,f^	1.8 ± 0.2 ^g,h^
		Tribe Chamaedoreeae								
6	*Chamaedorea microspadix*	1.6 ± 0.2 ^i,j,k,l^	11.9 ± 1.1 ^a^	0.2 ± 0.0 ^c,d^	1.3 ± 0.9 ^c,d,e,f,g,h^	5.2 ± 0.5 ^d^	0.6 ± 0.0 ^h^	4.2 ± 0.6 ^g,h,i,j^	7.3 ± 0.3 ^b,c^	0.1 ± 0.0 ^i^
7	*C. oblongata*	0.1 ± 0.0 ^l^	0.2 ± 0.0 ^i,j^	n.d	0.4 ± 0.0 ^j,k,l^	0.6 ± 0.0 ^k,l,m^	0.3 ± 0.0 ^h^	0.1 ± 0.0 ^l^	1.3 ± 0.0 ^i,j,k,l^	2.4 ± 0.3 ^g^
		Tribe Cocoseae								
8	*Butia capitata*	0.2 ± 0.0 ^l^	0.5 ± 0.0 ^f,g,h,i^	1.1 ± 0.1 ^c,d^	0.7 ± 0.0 ^g,h,i,j,k,l^	0.8 ± 0.3 ^j,k,l,m^	1.7 ± 0.3 ^h^	3.2 ± 0.2 ^h,i,j,k,l^	2.5 ± 0.3 ^g,h,i,j^	0.9 ± 0.1 ^h,i^
9	*Cocos nucifera*	0.4 ± 0.0 ^l,k^	0.9 ± 0.0 ^e,f,g,h,i^	1.7 ± 0.1 ^b,c,d^	1.9 ± 0.0 ^c,d^	0.7 ± 0.0 ^k,l,m^	1.3 ± 0.2 ^h^	0.9 ± 0.0 ^j,k,l^	3.2 ± 0.2 ^e,f,g,h^	0.4 ± 0.0 ^i^
10	*Syagrus romanzofianna*	0.3 ± 0.0 ^l^	0.6 ± 0.0 ^f,g,h,i^	0.1 ± 0.0 ^d^	0.5 ± 0.0 ^i,j,k,l^	n.d	4.9 ± 0.3 ^g^	11.3 ± 0.3 ^c,d^	0.4 ± 0.0 ^m^	14.8 ± 0.1 ^b,c^
		Subfamily Coryphoideae								
		Tribe Caryoteae								
11	*Arenga engleri*	0.8 ± 0.0 ^k,l^	0.3 ± 0.0 ^h,i,j^	1.4 ± 0.1 ^b,c,d^	1.0 ± 0.2 ^e,f,g,h,i,j^	13.1 ± 0.9 ^b^	0.2 ± 0.0 ^h^	n.d	0.1 ± 0.0 ^m^	11.6 ± 0.9 ^d^
		Tribe Phoeniceae								
12	*Phoenix canariensis*	8.6 ± 1.3 ^d^	0.4 ± 0.0 ^g,h,i,j^	n.d	1.7 ± 0.4 ^c,d,e^	0.1 ± 0.0 ^m^	7.1 ± 0.9 ^e,f^	8.5 ± 2.2 ^c,d,e,f^	0.3 ± 0.0 ^m^	5.7 ± 0.4 ^f^
13A	*P. dactylifera* var. *Deglet Nour*	1.5 ± 0.3 ^i,j,k,l^	0.3 ± 0.0 ^h,i,j^	n.d	0.4 ± 0.0 ^g,k,l^	22.0 ± 1.0 ^a^	4.4 ± 0.1 ^g^	n.d	n.d	6.2 ± 0.3 ^f^
13B	*P. dactylifera* var. *Medjool*	1.8 ± 0.2 ^i,j,k,l^	0.2 ± 0.0 ^i,j^	n.d	1.9 ± 0.2 ^c,d^	1.3 ± 0.1 ^h,i,j,k,l,m^	0.9 ± 0.1 ^h^	n.d	1.2 ± 0.0 ^j,k,l,m^	2.1 ± 0.3 ^g,h^
14	*P. reclinata*	0.9 ± 0.0 ^k,l^	n.d	n.d	0.9 ± 0.0 ^f,g,h,i,j,k^	3.2 ± 0.0 ^e^	0.1 ± 0.0 ^h^	n.d	n.d	0.3 ± 0.0 ^i^
		Tribe Sabaleae								
15	*Sabal bermudana*	6.9 ± 0.7 ^d,e^	0.3 ± 0.0 ^h,i,j^	n.d	0.7 ± 0.1 ^g,h,i,j,k,l^	2.4 ± 1.2 ^e,f,g,h^	8.5 ± 0.7 ^c,d,e^	8.9 ± 0.4 ^c,d,e,f^	8.5 ± 0.9 ^a,b^	n.d
16	*S. minor*	3.3 ± 0.0 ^g,h,i^	0.1 ± 0.0 ^j^	0.7 ± 0.0 ^c,d^	1.4 ± 0.1 ^c,d,e,f,g^	3.0 ± 0.1 ^e,f^	9.3 ± 0.6 ^c,d^	15.1 ± 1.5 ^b^	4.1 ± 0.1 ^e^	13.8 ± 0.1 ^c^
17	*S. palmetto*	16.2 ± 2.9 ^b^	0.2± 0.0 ^i,j^	1.9 ± 0.2 ^c,b,d^	1.0 ± 0.0 ^e,f,g,h,i,j^	7.2 ± 0.2 ^c^	15.7 ± 0.2 ^a^	33.5 ± 2.9 ^a^	1.9 ± 0.1 ^h,i,j,k,l^	17.1 ± 0.4 ^a^
18	*S. domingensis*	1.1 ± 0.0 ^j,k,l^	0.9 ± 0.0 ^e,f,g,h,i^	0.4 ± 0.0 ^c,d^	1.6 ± 0.1 ^c,d,e,f^	n.d	5.7 ± 0.3 ^f,g^	4.1 ± 0.3 ^g,h,i,j^	n.d	n.d
		Tribe Trachycarpeae								
19A	*Chamaerops humilis*	12.9 ± 1.3 ^c^	2.3 ± 0.4 ^d^	n.d	1.3 ± 0.0 ^c,d,e,f,g,h^	2.4 ± 0.1 ^e,f,g,h^	11.1 ± 0.7 ^b^	6.5 ± 0.5 ^fgh^	9.8 ± 1.2 ^a^	n.d
19B	*C. humilis*	19.9 ± 0.8 ^a^	2.2 ± 0.2 ^d^	0.4 ± 0.0 ^c,d^	1.4 ± 0.1 ^c,d,e,f,g^	2.3 ± 0.0 ^e,f,g,h,i^	16.1 ± 2.5 ^a^	8.4 ± 0.4 ^d,e,f^	1.0 ± 0.0 ^k,l,m^	5.6 ± 0.4 ^f^
19C	*C. humilis*	5.6 ± 0.4 ^e,f^	1.0 ± 0.0 ^e,f,g,h^	2.3 ± 2.8 ^b,c^	2.0 ± 0.1 ^c^	1.6 ± 0.1 ^g,h,i,j,k^	4.9 ± 0.2 ^g^	7.4 ± 0.8 ^e,f,g^	7.4 ± 0.3 ^b,c^	5.2 ± 0.3 ^f^
19D	*C. humilis*	5.7 ± 0.3 ^e,f^	1.4 ± 0.1 ^e^	0.6 ± 0.0 ^c,d^	1.2 ± 0.1 ^d,e,f,g,h,i^	2.1 ± 0.8 ^e,f,g,h,i^	1.5 ± 0.0 ^h^	10.3 ± 1.9 ^c,d,e^	2.6 ± 0.3 ^f,g,h,i^	8.8 ± 0.5 ^e^
20	*Livistona chinensis*	1.0 ± 0.0 ^j,k,l^	9.7 ± 0.7 ^b^	4.3 ± 0.5 ^a^	0.6 ± 0.0 ^h,i,j,k,l^	2.0 ± 0.0 ^e,f,g,h,i,j^	6.9 ± 0.3 ^e,f^	10.5 ± 3.1 ^c,d,e^	5.9 ± 0.3 ^d^	n.d
21	*L. fulva*	1.2 ± 0.0 ^j,k,l^	6.3 ± 0.1 ^c^	3.1 ± 0.1 ^b^	1.1 ± 0.1 ^e,f,g,h,i,j^	1.7 ± 0.0 ^g,h,i,j,k^	1.4 ± 0.1 ^h^	3.6 ± 0.1 ^h,i,j,k^	3.5 ± 0.2 ^e,f,g^	n.d
22	*L. saribus*	0.5 ± 0.0 ^k,l^	0.2 ± 0.0 ^i,j^	0.1 ± 0.0 ^d^	0.1 ± 0.0 ^l^	2.0 ± 0.0 ^e,f,g,h,i,j^	1.7 ± 0.0 ^h^	0.6 ± 0.0 ^k,l^	2.3 ± 0.1 ^g,h,i,j,k^	n.d
23	*Trachycarpus fortunei*	5.0 ± 0.2 ^e,f,g^	0.4 ± 0.0 ^g,h,i,j^	1.6 ± 0.2 ^b,c,d^	0.2 ± 0 ^l^	1.1 ± 0.1 ^i,j,k,l,m^	0.2 ± 0.0 ^h^	0.5 ± 0.0 ^k,l^	0.4 ± 0.0 ^m^	n.d
24	*Washingtonia robusta*	2.9 ± 0.2 ^h,i,j^	0.1 ± 0.0 ^j^	0.7 ± 0.1 ^c,d^	1.9 ± 0.3 ^c,d^	1.4 ± 0.0 ^h,i,j,k,l^	0.7 ± 0.0 ^h^	0.5 ± 0.0 ^k,l^	0.5 ± 0.0 ^m^	n.d
Code	Species	3,4-Dihydroxyhydrocinnamic Acid	Chlorogenic Acid	Caffeic Acid	(-)-Catechin	Syringic Acid	Dactylifric Acid^c^	*trans*-Coumaric Acid	Ferulic Acid	Sinapic Acid
	*Retention time (min)*	*30.56*	*35.21*	*36.32*	*38.53*	*39.10*	*42.76*	*44.64*	*47.72*	*49.38*
		Subfamily Arecoideae								
		Tribe Areceae								
1	*Archontophoenix cunninghamiana*	0.8 ± 0.0 ^b,c^	1.8 ± 0.3 ^a^	27.9 ± 1.6 ^d^	11.6 ± 1.4 ^g^	1.6 ± 0.2 ^g,h,i,j^	1.8 ± 0.0 ^j^	0.4 ± 0.0 ^f,g^	8.5 ± 0.6 ^a^	0.5 ± 0.0 ^f,g^
2	*Chambeyronia macrocarpa*	2.1 ± 0.3 ^a^	0.9 ± 0.2 ^c,d,e^	38.5 ± 0.9 ^c^	0.8 ± 0.0 ^j,k^	16.1 ± 1.0 ^b^	0.6 ± 0.0 ^j^	2.6 ± 2.3 ^d^	1.4 ± 0.0 ^h,i,j,k^	n.d
3	*Dypsis lutescens*	n.d	0.9 ± 0.1 ^c,d,e^	53.5 ± 3.6 ^b^	1.4 ± 0.2 ^i,j,k^	0.6 ± 0.1 ^i,j^	0.4 ± 0.0 ^j^	0.6 ± 0.0 ^f,g^	2.4 ± 0.1 ^e^	0.4 ± 0.1 ^f,g^
4	*Howea belmoreana*	0.1 ± 0.0 ^d^	0.4 ± 0.2 ^f,g,h^	1.2 ± 0.0 ^h,i,j,k^	17.1 ± 0.4 ^d^	0.1 ± 0.0 ^j^	1.8 ± 0.2 ^j^	0.4 ± 0.0 ^f,g^	2.2 ± 0.2 ^e,f^	7.7 ± 0.9 ^b^
5A	*H. forsteriana*	n.d	0.1 ± 0.1 ^h^	0.4 ± 0.0 ^j,k^	3.0 ± 0.1 ^h,i,j,k^	12.8 ± 0.9 ^c,d^	0.4 ± 0.0 ^j^	0.8 ± 0.1 ^e,f,g^	5.8 ± 0.3 ^b^	0.1 ± 0.0 ^g^
5B	*H. forsteriana*	0.1 ± 0.0 ^d^	0.2 ± 0.0 ^h^	2.5 ± 0.1 ^h,i,j,k^	1.0 ± 0.1 ^j,k^	2.6 ± 0.3 ^f,g,h^	1.0 ± 0.0 ^j^	0.7 ± 0.1 ^f,g^	4.5 ± 0.6 ^c^	0.5 ± 0.1 ^f,g^
		Tribe Chamaedoreeae								
6	*Chamaedorea microspadix*	2.3 ± 0.0 ^a^	0.2 ± 0.0^h^	0.9 ± 0.0 ^i,j,k^	3.1 ± 0.4 ^h,i,j,k^	0.3 ± 0.0 ^i,j^	4.3 ± 0.2 ^i,j^	0.6 ± 0.0 ^f,g^	0.2 ± 0.0 ^p^	0.5 ± 0.0 ^f,g^
7	*C. oblongata*	1.0 ± 0.0 ^b^	0.1 ± 0.0^h^	0.4 ± 0.0 ^j,k^	28.9 ± 0.0 ^a^	1.3 ± 0.2 ^h,i,j^	0.9 ± 0.0 ^j^	1.0 ± 0.0 ^e,f,g^	0.3 ± 0.0 ^o,p^	0.2 ± 0.0 ^f,g^
		Tribe Cocoseae								
8	*Butia capitata*	n.d	0.3 ± 0.0 ^g,h^	0.6 ± 0.0 ^i,j,k^	13.5 ± 0.7 ^f,g^	2.7 ± 0.2 ^f,g,h^	3.0 ± 0.2 ^i,j^	1.9 ± 0.2 ^d,e,f^	5.0 ± 0.3 ^c^	1.2 ± 0.1 ^d,e,f^
9	*Cocos nucifera*	n.d	1.8 ± 0.3 ^a^	0.1 ± 0.0 ^k^	17.9 ± 0.5 ^c,d^	1.5 ± 0.3 ^g,h,i,j^	1.9 ± 0.2 ^j^	0.9± 0.0 ^e,f,g^	1.8 ± 0.0 ^e,f,g,h^	0.9 ± 0.0 ^e,f,g^
10	*Syagrus romanzofianna*	n.d	0.8 ± 0.0 ^d,e,f^	n.d	15.1 ± 1.3 ^d,e^	3.2 ± 0.2 ^f,g^	2.0 ± 0.2 ^i,j^	0.1 ± 0.0 ^g^	4.5 ± 0.3^c^	1.4 ± 0.1 ^c,d,e,f^
		Subfamily Coryphoideae								
		Tribe Caryoteae								
11	*Arenga engleri*	n.d	0.8 ± 0.2 ^c,c,d,e,f^	3.7 ± 0.9 ^h,i^	20.1 ± 2.5 ^c^	8.1 ± 1.2 ^e^	1.6 ± 0.2 ^j^	1.3 ± 0.0 ^d,e,f,g^	0.6 ± 0.0 ^m,n,o,p^	0.5 ± 0.0 ^f,g^
		Tribe Phoeniceae								
12	*Phoenix canariensis*	0.2 ± 0.0 ^d^	0.5 ± 0.0 ^e,f,g,h^	16.2 ± 2.0 ^e^	4.2 ± 0.2 ^h,i^	3.8 ± 0.7 ^f^	0.4 ± 0.0 ^j^	28.1 ± 1.3 ^a^	3.2 ± 0.2 ^d^	2.5 ± 0.3 ^c^
13A	*P. dactylifera* var. *Deglet Nour*	0.3 ± 0.0 ^d^	0.2 ± 0.0 ^h^	1.2 ± 0.0 ^h,i,j,k^	25.1 ± 3.5 ^b^	11.4 ± 0.2 ^d^	27.4 ± 3.1 ^d,e^	11.8 ± 0.3 ^c^	0.9 ± 0.0 ^j,k,l,m,n,o^	8.1 ± 0.6 ^b^
13B	*P. dactylifera* var. *Medjool*	0.7 ± 0.0 ^c^	1.6 ± 0.2 ^a,b^	2.6 ± 1.1 ^h,i,j,k^	20.7 ± 1.9^c^	19.3 ± 2.1 ^a^	31.2 ± 1.1 ^c,d^	27.2 ± 0.0 ^a^	1.7 ± 0.3 ^f,g,h,i^	0.9 ± 0.0 ^e,f,g^
14	*P. reclinata*	0.2 ± 0.0 ^d^	0.7 ± 0.0 ^d,e,f,g^	1.9 ± 0.2 ^h,i,j,k^	12.5 ± 1.4 ^f,g^	13.5 ± 0.3 ^c^	29.1 ± 0.1 ^d,e^	20.9 ± 0.8 ^b^	0.2 ± 0.0 ^p^	0.5 ± 0.1 ^f,g^
		Tribe Sabaleae								
15	*Sabal bermudana*	n.d	1.6 ± 0.0 ^a,b^	4.2 ± 0.0 ^g,h^	4.8 ± 0.6 ^h^	2.4 ± 0.1 ^f,g,h^	79.7 ± 12.1 ^a^	0.6 ± 0.1 ^f,g^	n.d	1.8 ± 0.1 ^c,d,e^
16	*S. minor*	n.d	0.5 ± 0.0 ^e,f,g,h^	16.1 ± 0.6 ^e^	3.6 ± 0.5 ^h,i,j^	1.0 ± 0.1 ^h,i,j^	38.1 ± 3.1 ^b,c^	2.3 ± 0.1 ^d,e^	1.1 ± 0.2 ^i,j,k,l,m,n^	1.9 ± 0.3 ^c,d,e^
17	*S. palmetto*	n.d	0.4 ± 0.0 ^f,g,h^	15.1 ± 1.2 ^e^	1.5 ± 0.2 ^i,j,k^	1.5 ± 0.1 ^g,h,i,j^	22.3 ± 2.1 ^e,f^	0.3 ± 0.0 ^g^	1.2 ± 0.3 ^h,i,j,k,l,m^	n.d
18	*S. domingensis*	n.d	1.1 ± 0.0 ^c,d^	14.5 ± 1.5 ^e^	2.6 ± 0.3 ^h,i,j,k^	n.d	21.7 ± 0.1 ^e,f^	0.9 ± 0.0 ^e,f,g^	0.7 ± 0.0 ^l,m,n,o,p^	n.d
		Tribe Trachycarpeae								
19A	*Chamaerops humilis*	n.d	0.2 ± 0.0 ^h^	0.5 ± 0.0 ^j,k^	0.9 ± 0.1 ^j,k^	n.d	1.2 ± 0.1 ^j^	0.9 ± 0.2 ^e,f,g^	1.3 ± 0.1 ^h,i,j,k,l^	n.d
19B	*C. humilis*	n.d	0.1 ± 0.1 ^h^	0.4 ± 0.0 ^j,k^	1.4 ± 0.2 ^i,j,k^	n.d	0.1 ± 0.0 ^j^	0.1 ± 0.0 ^g^	1.5 ± 0.2 ^g,h,i,j^	15.1 ± 1.7 ^a^
19C	*C. humilis*	n.d	0.3 ± 0.0 ^gh^	0.8 ± 0.1 ^i,j,k^	3.0 ± 0.2 ^h,i,j,k^	n.d	0.5 ± 0.0 ^j^	0.3 ± 0.0 ^g^	0.6 ± 0.2 ^m,n,o,p^	0.2 ± 0.0 ^f,g^
19D	*C. humilis*	n.d	1.2 ± 0.6 ^b,c^	2.2 ± 0.1 ^h,i,j,k^	0.4 ± 0.0 ^k^	1.6 ± 0.3 ^g,h,i,j^	0.1 ± 0.0 ^j^	0.4 ± 0.0 ^f,g^	2.1 ± 0.2 ^e,f,g,^	2.4 ± 0.1 ^c,d^
20	*Livistona chinensis*	n.d	0.5 ± 0.0 ^e,f,g,h^	66.1 ± 2.3 ^a^	2.6 ± 0.2 ^h,i,j,k^	1.3 ± 0.0 ^h,i,j^	39.1 ± 0.4 ^b^	0.4 ± 0.0 ^f,g^	0.8 ± 0.0 ^k,l,m,n,o,p^	0.2 ± 0.0 ^f,g^
21	*L. fulva*	n.d	0.3 ± 0.1 ^g,h^	8.0 ± 0.3 ^f^	1.3 ± 0.2 ^i,j,k^	2.0 ± 0.0 ^g,h,i^	12.4 ± 0.6 ^g,h^	0.2 ± 0.0 ^g^	0.5 ± 0.0 ^n,o,p^	n.d
22	*L. saribus*	n.d	0.5 ± 0.1 ^e,f,g,h^	8.9 ± 0.1 ^f^	1.5 ± 0.3 ^i,j,k^	0.2 ± 0.0 ^j^	18.0 ± 0.1 ^f,g^	1.0 ± 0.0 ^e,f,g^	0.2 ± 0.0 ^p^	n.d
23	*Trachycarpus fortunei*	n.d	0.1 ± 0.0 ^h^	3.4 ± 0.3 ^h,i,j^	2.1 ± 0.3 ^h,i,j,k^	1.8 ± 0.3 ^g,h,i,j^	12.8 ± 0.3 ^g,h^	0.5 ± 0.0 ^f,g^	0.6 ± 0.0 ^m,n,o,p^	n.d
24	*Washingtonia robusta*	n.d	0.4 ± 0.0 ^f,g,h^	7.3 ± 0.4 ^f,g^	0.4 ± 0.0 ^k^	1.6 ± 0.6 ^g,h,i,j^	9.0 ± 0.2 ^h,i^	1.0 ± 0.0 ^e,f,g^	0.3 ± 0.0 ^o,p^	0.4 ± 0.0 ^f,g^
Code	Species	Eriodictyol ^d^	Rutin	Rosmarinic Acid	2-OH-4-methoxybenzoic Acid	Naringenin	Quercetin	Luteolin	Kaempferol	Total Phenolics (mg/100 g)
	*Retention time (min)*	*54.26*	*57.35*	*58.65*	*62.87*	*65.96*	*68.76*	*72.28*	*76.73*	
		Subfamily Arecoideae								
		Tribe Areceae								
1	*Archontophoenix cunninghamiana*	2.9 ± 0.2 ^e,f,g^	22.9 ± 3.3 ^d^	41.8 ± 1.6 ^a^	5.1 ± 0.4 ^d^	9.5 ± 1.6 ^a^	1.8 ± 0.1 ^d^	3.1 ± 0.5 ^e^	22.8 ± 2.2 ^a^	197.5 ± 5.6 ^b^
2	*Chambeyronia macrocarpa*	1.4 ± 0.3 ^h,i,j,k,l^	1.7 ± 0.1 ^k,l,m^	6.6 ± 0.6 ^c^	3.7 ± 0.2 ^e^	0.6 ± 0.0 ^e,f^	0.3 ± 0.0 ^i,j,k^	0.4 ± 0.0 ^f^	0.1 ± 0.0 ^e^	88.2 ± 2.9 ^f,g^
3	*Dypsis lutescens*	3.7 ± 0.3 ^e^	2.4 ± 0.2 ^j,k,l,m^	3.0 ± 0.0 ^d,e,f^	1.7 ± 0.0 ^h,i^	0.1 ± 0.0 ^f^	0.7 ± 0.1 ^f,g,h^	0.1 ± 0.0 ^f^	0.7 ± 0.0 ^d,e^	120.7 ± 4.1 ^e^
4	*Howea belmoreana*	6.3 ± 0.5 ^c^	26.1 ± 2.3 ^d^	0.1 ± 0.0 ^j^	0.3 ± 0.0 ^j^	0.7 ± 0.0 ^e,f^	1.7 ± 0.0 ^d^	0.2 ± 0.0 ^f^	0.5 ± 0.0 ^e^	96.1 ± 2.8 ^f^
5A	*H. forsteriana*	20.7 ± 1.5 ^a^	15.3 ± 1.4 ^e,f^	0.3 ± 0.0 ^i,j^	0.3 ± 0.0 ^j^	0.4 ± 0.0 ^e,f^	0.7 ± 0.0 ^f,g,h^	0.4 ± 0.0 ^f^	0.3 ± 0.0 ^e^	77.0 ± 2.3 ^h^
5B	*H. forsteriana*	9.0 ± 0.8 ^b^	35.4 ± 2.9 ^c^	0.4 ± 0.0 ^h,i,j^	0.9 ± 0.0 ^i,j^	0.2 ± 0.0 ^f^	0.4 ± 0.0 ^h,i,j,k^	0.3 ± 0.0 ^f^	0.5 ± 0.0 ^e^	75.3 ± 3.1 ^h^
		Tribe Chamaedoreeae								
6	*Chamaedorea microspadix*	0.5 ± 0 ^k,l,m^	1.7 ± 0.2 ^k,l,m^	1.4 ± 0.3 ^g,h,i,j^	2.3 ± 0.5 ^f,g,h^	2.3 ± 0.1 ^c,d^	0.3 ± 0.0 ^i,j,k^	0.6 ± 0.0 ^f^	0.1 ± 0.0 ^e^	40.3 ± 1.4 ^n,o^
7	*C. oblongata*	0.2 ± 0 ^m^	0.3 ± 0.0 ^m^	1.9 ± 0.1 ^e,f,g,h^	n.d	1.3 ± 0.2 ^d,e^	0.2 ± 0.0 ^j,k^	0.4 ± 0.0 ^f^	0.2 ± 0.0 ^e^	43.7 ± 0.4 ^lmno^
		Tribe Cocoseae								
8	*Butia capitata*	3.6 ± 0.3 ^e^	3.5 ± 0.1 ^j,k,l,m^	0.4 ± 0.0 ^h,i,j^	0.9 ± 0.0 ^i,j^	0.9 ± 0.0 ^e,f^	0.6 ± 0.0 ^f,g,h,i^	1.2 ± 0.3 ^e,f^	0.2 ± 0.0 ^e^	49.3 ± 1.1 ^k,l,m,n,o^
9	*Cocos nucifera*	1.9 ± 0.4 ^g,h,i^	5.9 ± 0.4 ^i,j^	1.1 ± 0.0 ^h,i,j^	0.2 ± 0.0 ^j^	0.3 ± 0.0 ^e,f^	5.0 ± 0.3 ^a^	0.4 ± 0.0 ^f^	4.5 ± 0.0 ^b^	54.5 ± 1.0 ^j,k,l^
10	*Syagrus romanzofianna*	2.9 ± 0.3 ^e,f,g^	2.7 ± 0.0 ^j,k,l,m^	0.5 ± 0.0 ^h,i,j^	1.8 ± 0.2 ^g,h,i^	4.3 ± 0.4 ^b^	2.3 ± 0.2 ^c^	1.4 ± 0.2 ^e,f^	2.1 ± 0.3 ^c^	77.0 ± 1.6 ^h^
		Subfamily Coryphoideae								
		Tribe Caryoteae								
11	*Arenga engleri*	3.1 ± 0.3 ^e,f^	11.8 ± 1.2 ^f,g,h^	3.4 ± 0.4 ^d,e^	11.9 ± 0.8 ^a^	n.d	0.2 ± 0.0 ^j,k^	0.3 ± 0.0 ^f^	0.2 ± 0.0 ^e^	93.6 ± 3.5 ^f^
		Tribe Phoeniceae								
12	*Phoenix canariensis*	0.7 ± 0.0 ^j,k,l,m^	2.3 ± 0.2 ^j,k,l,m^	0.3 ± 0.0 ^i,j^	n.d	n.d	0.6 ± 0.0 ^f,g,h,i^	1.7 ± 0.1 ^e,f^	0.3 ± 0.0 ^e^	88.4 ± 3.5 ^f,g^
13A	*P. dactylifera* var. *Deglet Nour*	1.0 ± 0.1 ^i,j,k,l,m^	4.6 ± 0.8 ^j,k,l^	3.5 ± 0.9 ^d^	1.2 ± 0.2 ^i,j^	3.0 ± 0.2 ^c^	0.3 ± 0.0 ^i,j,k^	0.5 ± 0.0 ^f^	0.6 ± 0.0 ^e^	134.1 ± 5.0 ^d^
13B	*P. dactylifera* var. *Medjool*	0.4 ± 0.0 ^l,m^	11.9 ± 1.9 ^f,g,h^	14.3 ± 1.7 ^b^	0.4 ± 0.0 ^j^	0.5 ± 0.0 ^e,f^	0.9 ± 0.0 ^f^	0.7 ± 0.0 ^f^	1.9 ± 0.3 ^c,d^	144.3 ± 4.2 ^d^
14	*P. reclinata*	0.6 ± 0.0 ^k,l,m^	0.6 ± 0.0 ^m^	1.8 ± 0.0 ^f,g,h,i^	0.2 ± 0.0 ^j^	0.1 ± 0.0 ^f^	0.5 ± 0.0 ^g,h,i,j^	0.3 ± 0.0 ^f^	0.9 ± 0.0 ^c,d,e^	89.0 ± 1.7 ^f,g^
		Tribe Sabaleae								
15	*Sabal bermudana*	2.9 ± 0.1 ^e,f,g^	10.6 ± 0.9 ^g,h^	3.5 ± 0.6 ^d^	1.2 ± 0.3 ^i,j^	0.3 ± 0.0 ^e,f^	1.7 ± 0.2 ^d^	14.5 ± 0.0 ^c^	0.4 ± 0.0 ^e^	159.2 ± 12.3 ^c^
16	*S. minor*	3.4 ± 0.6 ^e^	49.6 ± 1.9 ^b^	5.6 ± 0.1 ^c^	2.8 ± 0.7 ^e,f,g^	2.8 ± 0.3 ^c^	0.8 ± 0.1 ^f,g^	24.6 ± 1.5 ^b^	1.1 ± 0.2 ^c,d,e^	202.0 ± 4.5 ^b^
17	*S. palmetto*	3.2 ± 0.3 ^e,f^	58.2 ± 2.1 ^a^	5.7 ± 0.0 ^c^	3.0 ± 0.2 ^e,f^	0.3 ± 0.0 ^e,f^	0.9 ± 0.0 ^f^	55.6 ± 3.8 ^a^	0.4 ± 0.0 ^e^	246.0 ± 5.8 ^a^
18	*S. domingensis*	2.2 ± 0.3 ^f,g,h^	14.3 ± 1.2 ^e,f,g^	3.4 ± 0.6 ^d,e^	5.6 ± 0.4 ^d^	0.1 ± 0.0 ^f^	1.3 ± 0.0 ^e^	11.4 ± 0.1 ^d^	1.3 ± 0.2 ^c,d,e^	91.5 ± 2.2 ^f^
		Tribe Trachycarpeae								
19A	*Chamaerops humilis*	1.8 ± 0.1 ^g,h,i,j^	4.9 ± 0.0 ^i,j,k^	1.7 ± 0.1 ^f,g,h,i^	12.0 ± 0.3 ^a^	0.2 ± 0.0 ^f^	0.3 ± 0.0 ^i,j,k^	0.2 ± 0.0 ^f^	0.1 ± 0.0 ^e^	57.3 ± 1.5 ^i,j,k^
19B	*C. humilis*	0.2 ± 0.0 ^m^	13.5 ± 0.7 ^e,f,g^	5.2 ± 0.4 ^c^	7.2 ± 0.4 ^c^	0.3 ± 0.0 ^e,f^	0.4 ± 0.0 ^h,i,j,k^	0.2 ± 0.0 ^f^	0.3 ± 0.0 ^e^	80.8 ± 3.2 ^g,h^
19C	*C. humilis*	0.5 ± 0.0 ^k,l,m^	8.5 ± 0.6 ^h,i^	2.8 ± 0.3 ^d,e,f,g^	2.4 ± 0.1 ^f,g,h^	0.6 ± 0.0 ^e,f^	0.1 ± 0.0 ^k^	0.7 ± 0.0 ^f^	0.6 ± 0.0 ^e^	50.4 ± 1.2 ^k,l,m,n^
19D	*C. humilis*	0.7 ± 0.0 ^j,k,l,m^	5.5 ± 0.5 ^i,j^	1.4 ± 0.3 ^g,h,i,j^	7.0 ± 0.5 ^c^	0.2 ± 0.0 ^f^	0.2 ± 0.0 ^j,k^	0.1 ± 0.0 ^f^	0.2 ± 0.0 ^e^	52.2 ± 2.4 ^j,k,l,m^
20	*Livistona chinensis*	1.6 ± 0.2 ^h,i,j,k^	0.1 ± 0.0 ^m^	0.8 ± 0.0 ^h,i,j^	n.d	n.d	0.3 ± 0.0 ^i,j,k^	0.2 ± 0.0 ^f^	0.3 ± 0.0 ^e^	140.2 ± 3.9 ^d^
21	*L. fulva*	0.6 ± 0.3 ^k,l,m^	0.6 ± 0.0 ^m^	0.9 ± 0.1 ^h,i,j^	n.d	n.d	0.4 ± 0.0 ^h,i,j,k^	0.1 ± 0.0 ^f^	0.4 ± 0.0 ^e^	39.0 ± 0.8 ^o^
22	*L. saribus*	1.9 ± 0.4 ^g,h,i^	0.9 ± 0.0 ^l,m^	1.8 ± 0.2 ^f,g,h,i^	n.d	n.d	0.6 ± 0.0 ^f,g,h,i^	0.4 ± 0.0 ^f^	0.7 ± 0.0 ^d,e^	43.3 ± 0.6 ^m,n,o^
23	*Trachycarpus fortunei*	9.2 ± 0.4 ^b^	14.5 ± 0.5 ^e,f^	0.7 ± 0.0 ^h,i,j^	12.9 ± 0.7 ^a^	1.3 ± 0.2 ^d,e^	1.5 ± 0.3 ^d,e^	0.3 ± 0.0 ^f^	1.2 ± 0.0 ^c,d,e^	65.3 ± 1.2 ^i^
24	*Washingtonia robusta*	5.0 ± 0.3 ^d^	15.8 ± 2.1 ^e^	1.2 ± 0.2 ^h,i,j^	9.3 ± 0.5 ^b^	0.9 ± 0.0 ^e,f^	3.7 ± 0.4 ^b^	0.1 ± 0.0 ^f^	0.5 ± 0.0 ^e^	61.9 ± 2.4 ^i,j^

^a^ Results are the mean value ± standard deviation (n = 3). Data sharing the same superscript letter in each column are not significantly different (*p* < 0.05) according to one-way ANOVA followed by Duncan’s test; ^b^ gallic acid equivalents; ^c^ syringic acid equivalents; ^d^ quercetin equivalents; n.d.: not detected.

**Table 3 plants-12-00226-t003:** Phenolic compounds detected by LC-MS in seeds of the analyzed Arecaceae species.

Code	Species	Resveratrol	Piceatannol	Pinocembrin	Genistein	Apigenin	Phloretin	Luteolin	Sakuranetin	Cianidin	Epicatechin (-)
	*m/z* precursor ion	227.07137	243.06628	255.06628	271.0601	271.0601	275.0914	285.04046	285.07685	287.05501	291.08631
Subfamily Arecoideae											
Tribe Areceae											
1	*Archontophoenix cunninghamiana*	-	-	-	-	-	-	+	-	-	+
2	*Chambeyronia macrocarpa*	-	-	-	-	-	-	+	-	-	+
3	*Dypsis lutescens*	-	-	-	-	-	-	+	-	-	-
4	*Howea belmoreana*	-	-	-	-	-	-	-	-	-	+
5A	*H. forsteriana*	-	-	-	-	-	-	-	-	-	+
5B	*H. forsteriana*	-	-	-	-	-	-	-	-	-	-
Tribe Chamaedoreeae											
6	*Chamaedorea microspadix*	-	-	-	-	-	-	-	-	-	-
7	*C. oblongata*	-	-	-	-	-	-	+	-	-	-
Tribe Cocoseae											
8	*Butia capitata*	-	-	-	-	-	-	-	-	-	+
9	*Cocos nucifera*	-	-	-	-	-	-	-	-	-	-
10	*Syagrus romanzofianna*	+	+	-	-	-	+	-	+	-	+
Subfamily Coryphoideae											
Tribe Caryoteae											
11	*Arenga engleri*	**-**	**-**	**-**	**-**	**-**	**+**	**-**	**-**	**-**	**+**
Tribe Phoeniceae											
12	*Phoenix canariensis*	-	-	-	-	-	-	+	-	-	+
											-
13A	*P. dactylifera* var. *Deglet Nour*	-	-	-	-	-	-	-	-	-	+
13B	*P. dactylifera* var. *Medjool*	-	-	-	-	-	-	+	-	-	+
14	*P. reclinata*	-	-	-	+	+	-	-	-	-	+
Tribe *Sabaleae*											
15	*Sabal bermudana*	-	-	-	-	-	-	+	-	-	+
16	*S. minor*	-	-	-	-	-	-	+	-	-	+
17	*S. palmetto*	-	-	-	-	-	-	+	-	-	+
18	*S. domingensis*	-	-	-	+	+	-	-	-	-	+
Tribe Trachycarpeae											
19A	*Chamaerops humilis*	-	-	-	-	-	-	-	-	-	+
19B	*C. humilis*	-	-	-	-	-	-	+	-	-	+
19C	*C. humilis*	-	-	-	-	-	-	-	-	-	-
19D	*C. humilis*	-	-	-	-	-	-	-	-	-	-
20	*Livistona chinensis*	-	-	-	-	-	-	-	-	-	+
21	*L. fulva*	-	+	-	-	-	+	-	-	+	+
22	*L. saribus*	-	-	+	+	+	-	-	-	-	+
23	*Trachycarpus fortunei*	-	-	-	-	-	-	-	-	-	+
24	*Washingtonia robusta*	-	-	-	-	-	-	-	-	-	-
Code	Species	Delphinidine	Epigallocatechin (-)	Gallocatechin (-)	Petunidine	Bilobalide	Malvidine	Ferulic Acid Hexoside	Piceid (resveratrol-3-*O*-beta-D-glucopyranoside)	Apigenin-6-C-glucoside (Isovitexin)	Phloridzin
	*m/z* precursor ion	303.04993	305.06668	307.08123	317.06558	325.09289	331.08123	355.10346	391.13874	431.09837	435.12967
Subfamily Arecoideae											
Tribe Areceae											
1	*Archontophoenix cunninghamiana*	-	-	-	-	-	-	+	-	-	-
2	*Chambeyronia macrocarpa*	-	-	-	-	-	-	+	-	-	-
3	*Dypsis lutescens*	-	-	-	-	-	-	+	-	-	-
4	*Howea belmoreana*	-	-	-	-	-	-	-	-	-	-
5A	*H. forsteriana*	-	-	-	-	-	-	+	-	-	-
5B	*H. forsteriana*	-	-	-	-	-	-	-	-	-	-
Tribe Chamaedoreeae											
6	*Chamaedorea microspadix*	-	-	-	-	-	-	-	-	-	-
7	*C. oblongata*	-	-	-	-	-	-	-	-	-	-
Tribe Cocoseae											
8	*Butia capitata*	-	-	-	-	-	-	-	-	-	-
9	*Cocos nucifera*	-	-	-	-	-	-	-	-	-	-
10	*Syagrus romanzofianna*	-	-	+	-	-	-	-	-	-	+
Subfamily Coryphoideae											
Tribe Caryoteae											
11	*Arenga engleri*	**-**	**-**	**+**	**-**	**-**	**-**	-	+	-	+
Tribe Phoeniceae											
12	*Phoenix canariensis*	-	-	-	-	-	-	-	-	-	-
13A	*P. dactylifera* var. *Deglet Nour*	-	-	-	-	-	-	-	-	-	-
13B	*P. dactylifera* var. *Medjool*	-	-	-	-	-	-	-	-	-	-
14	*P. reclinata*	-	-	-	-	+	-	-	-	-	+
Tribe Sabaleae											
15	*Sabal bermudana*	-	-	-	-	-	-	-	-	-	-
16	*S. minor*	-	-	-	-	-	-	-	-	-	-
17	*S. palmetto*	-	-	-	-	-	-	-	-	-	-
18	*S. domingensis*	+	+	+	+	+	-	-	-	+	-
Tribe Trachycarpeae											
19A	*Chamaerops humilis*	-	-	-	-	-	-	+	-	-	-
19B	*C. humilis*	-	-	-	-	-	-	+	-	-	-
19C	*C. humilis*	-	-	-	-	-	-	-	-	-	-
19D	*C. humilis*	-	-	-	-	-	-	-	-	-	-
20	*Livistona chinensis*	-	-	-	-	-	-	-	-	-	-
21	*L. fulva*	+	+	-	+	-	-	-	-	+	-
22	*L. saribus*	-	+	+	+	+	+	-	-	-	+
23	*Trachycarpus fortunei*	-	+	+	-	-	-	-	-	-	+
24	*Washingtonia robusta*	-	+	+	-	-	-	-	-	+	+
Code	Species	Quercetin-3-*O*-rhamnoside (Quercitrin)	Isorhamnetin-3-*O*-Glucoside	Lithospermic Acid	Procyanidin B1	Kaempferol-3-*O*-Rutinoside	Pelargonidine	Eriocitrin	Hesperidin	Delphinidin-3-*O*-(6-*O*-*p*-coumaroyl)-Glucoside	Isohamnetin-3-*O*-Rutinoside
	*m/z* precursor ion	447.09328	479.11840	537.10385	577.13515	593.15119	595.16575	595.16684	609.18249	611.13953	623.16176
Subfamily Arecoideae											
Tribe Areceae											
1	*Archontophoenix cunninghamiana*	-	-	-	-	-	-	-	-	-	-
2	*Chambeyronia macrocarpa*	-	-	-	-	-	-	-	-	-	-
3	*Dypsis lutescens*	-	-	-	-	-	-	-	-	-	-
4	*Howea belmoreana*	-	-	-	-	-	-	-	-	-	-
5A	*H. forsteriana*	-	-	-	-	-	-	-	-	-	-
5B	*H. forsteriana*	-	-	-	-	-	-	-	-	-	-
Tribe Chamaedoreeae											
6	*Chamaedorea microspadix*	-	-	-	-	-	-	-	-	-	-
7	*C. oblongata*	-	-	-	-	-	-	-	-	-	-
Tribe Cocoseae											
8	*Butia capitata*	-	-	-	-	-	-	-	-	-	-
9	*Cocos nucifera*	-	-	-	-	-	-	-	-	-	-
10	*Syagrus romanzofianna*	-	-	-	+	-	-	-	+	-	-
Subfamily Coryphoideae											
Tribe Caryoteae											
11	*Arenga engleri*	+	-	-	+	-	-	-	+	-	-
Tribe Phoeniceae											
12	*Phoenix canariensis*	-	-	-	+	-	-	-	-	-	-
13A	*P. dactylifera* var. *Deglet Nour*	-	-	-	-	-	-	-	-	-	-
13B	*P. dactylifera* var. *Medjool*	-	-	-	-	-	-	-	-	-	-
14	*P. reclinata*	-	-	-	-	+	+	-	-	-	-
Tribe Sabaleae											
15	*Sabal bermudana*	-	-	+	-	-	-	-	-	-	-
16	*S. minor*	-	-	-	-	-	-	-	-	-	-
17	*S. palmetto*	-	-	-	-	-	-	-	-	-	-
18	*S. domingensis*	-	-	-	+	-	-	-	-	-	-
Tribe Trachycarpeae											
19A	*Chamaerops humilis*	-	-	-	-	-	-	-	-	-	-
19B	*C. humilis*	-	-	-	-	-	-	-	-	-	-
19C	*C. humilis*	-	-	-	-	-	-	-	-	-	-
19D	*C. humilis*	-	-	-	-	-	-	-	-	-	-
20	*Livistona chinensis*	-	-	+	-	+	-	-	-	-	-
21	*L. fulva*	+	+	-	+	+	+	+	+	+	-
22	*L. saribus*	+	-	-	+	+	+	-	-	-	-
23	*Trachycarpus fortunei*	+	-	-	+	-	+	+	+	-	+
24	*Washingtonia robusta*	+	-	+	+	+	-	-	+	-	+

**Table 4 plants-12-00226-t004:** Identification, location, and date of collection of the Arecaceae seeds analyzed in this work.

Sample Code	Species	Common Name	Sample Location	Geographical Coordinates	Collection Date
	Subfamily Arecoideae				
	Tribe Areceae				
1	*Archontophoenix cunninghamiana* H. Wendl. & Drude	Bangalow palm	Botanic gardens of wood Rui Vieira, Portugal	32.662316, −16.894604	2020
2	*Chambeyronia macrocarpa* (Brongn.) Vieill. Ex Becc.	Red leaf palm	Botanic gardens of wood Rui Vieira, Portugal	32.662316, −16.894604	2020
3	*Dypsis lutescens (H. Wendl.) Beentje & J.Dransf.* (H. Wendl.) Beentje & J. Dransf. (Syn. *Chrysalidocarpus lutescens* H. Wendl.)	Yellow cane palm Yellow areca palm	Botanic gardens of wood Rui Vieira, Portugal	32.662316, −16.894604	2020
4	*Howea belmoreana* (C. Moore & F. Muell.) Becc.	Curly palm	Botanic gardens of wood Rui Vieira, Portugal	32.662316, −16.894604	2020
5A	*Howea forsteriana* Becc.	Kentia palm	Botanic gardens of wood Rui Vieira, Portugal	32.662316, −16.894604	2020
5B	*H. forsteriana* Bec	Kentia palm	Botanischer Garten Berlin−Dahlem 3550, Germany	52.456684, 13.304710	2020
	Tribe Chamaedoreeae				
6	*Chamaedorea microspadix* Burret	Hardy bamboo palm	Palm Society, San Leandro, California	37.727389, −122.180107	2021
7	*Chamaedorea oblongata* Mart. VU bk.	Caquib, palmilla, chate	Botanic garden of the University, Bulgaria	42.697102, 23.334565	2018
	Tribe Cocoseae				
8	*Butia capitata* (Mart.) Becc.	Butià, Jelly palm	Botanic gardens of wood Rui Vieira, Portugal	32.662316, −16.894604	2020
9	*Cocos nucifera* L.	Coconut palm	Malaga, Spain (Purchased)		2021
10	*Syagrus romanzoffiana* (Cham.) Glassman	Queen palm	University of Almería gardens, Spain	36.829694, −2.404185	2021
	Subfamily Coryphoideae				
	Tribe Caryoteae				
11	*Arenga engleri* Becc.	Taiwan sugar palm	University Botanic Garden of Valencia, Spain	39.475663, −0.386351	2021
	Tribe Phoeniceae				
12	*Phoenix canariensis* Chabaud	Canary Island date palm	Botanic gardens of wood Rui Vieira, Portugal	32.662316, −16.894604	2020
13A	*Phoenix dactylifera* L. var. *Deglet Nour*	Date palm	Algeria (Purchased)		2021
13B	*P. dactylifera* L. var. *Medjool*	Date palm	Spain (Purchased)		2021
14	*Phoenix reclinata* Jacq.	Wild date palm	University Botanic Garden of Valencia, Spain	39.475663, −0.386351	2021
	Tribe Sabaleae				
15	*Sabal bermudana* L. H. Bailey ex Knuth	Bermuda palmetto	University Botanic gardens, Bulgaria	42.697102, 23.334565	2018
18	*Sabal domingensis* Becc.	Hispaniola palmetto	University Botanic Garden of Valencia, Spain	39.475663, −0.386351	2021
16	*Sabal minor* (Jacq.) Pers.	Dwarf palmetto	University Botanic gardens, Bulgaria	42.697102, 23.334565	2019
17	*Sabal palmetto* (Walt.) Lodd.	Cabbage palm	Florida, Miami, Coral Gables, USA	25.294750, −76.188889	2021
	Tribe Trachycarpeae				
19A	*Chamaerops humilis* L.	Mediterranean fan palm	Bulgaria, University Botanic gardens	42.697102, 23.334565	2020
19B	*C. humilis* L.	Mediterranean fan palm	Bulgaria, University Botanic gardens	42.697102, 23.334565	2020
19C	*C. humilis* L.	Mediterranean fan palm	Pernambuco, Brazil	−969.12055107, −36.6077802	2021
19D	*C. humilis* L.	Mediterranean fan palm	El Toyo, Almería, Spain	36.836508, −2.326255	2021
20	*Livistona chinensis* (Jacq.) R.Br. ex Mart. sf.	Chinese fan palm	Bulgaria, University Botanic gardens	42.697102, 23.334565	2020
21	*Livistona fulva* Rodd	Blackdown fan palm	University Botanic Garden of Valencia, Spain	39.475663, −0.386351	2020
22	*Livistona saribus* (Lour.) Merr. ex A.Chev.	Taraw palm	University Botanic Garden of Valencia, Spain	39.475663, −0.386351	2022
23	*Trachycarpus fortunei* (Hook.) H.Wendl	Windmill palm	University Botanic Garden of Valencia, Spain	39.475663, −0.386351	2022
24	*Washingtonia robusta* H. Wendl.	Washington fan palm	University Botanic Garden of Valencia, Spain	39.475663, −0.386351	2021

## Data Availability

Not applicable.

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
