# Peer review of "Arecaceae Seeds Constitute a Healthy Source of Fatty Acids and Phenolic Compounds"

_plants, 2023, doi:10.3390/plants12020226_

Round 1

Reviewer 1 Report

1. Phenolic compounds and organic acids of seeds from Arecaceae species, the subject addressed is one of interest.

2. The subject is original because it addresses performance analysis techniques the to 23 phenolic compounds were identified and quantified by HPLC-DAD in the seeds from the screened species.

3. I consider it a complex article, of interest for the exploitation of these Arecaceae species.

Author Response

I deeply appreciate the reviewer's positive comments.

Reviewer 2 Report

Palm seeds are a valuable material for the food industry, especially those varieties that have a high level of medium-chain MCT acids are sought after. In the presented manuscript, the authors raised the issue of the content of fatty acids, polyphenolic compounds and their biological activity against the HT-29 cancer cell line. The work is edited correctly, although it requires some corrections.

- in line 92 add the number of tested varieties

In table 1.

add up the content of medium chain acids 8:0 and 10:0 and express them in % of total fatty acid content, similarly for long chain acids. Include the changes made in the discussion of the results. In the discussion of the results, rely on the results of statistical analysis. Please explain why in Table 2 a different font is used for Dactylfric acid and Eriodictyol.

Chapter 4.3

line 463 enter the duration of the extraction. This parameter was not given in the additional materials either.

Author Response

Answers (#) to reviewer comments (-)

- in line 92 add the number of tested varieties

# This suggestion has been carried out.

-In table 1.

add up the content of medium chain acids 8:0 and 10:0 and express them in % of total fatty acid content, similarly for long chain acids. Include the changes made in the discussion of the results. In the discussion of the results, rely on the results of statistical analysis. Please explain why in Table 2 a different font is used for Dactylfric acid and Eriodictyol.

# Please, note that Table 1 details the content of 8:0 and 10:0 in % of total fatty acids.

Regarding the results of the statistical analysis, for each tribe/species we have selected the most significant results worthy to be discussed, and the text of the manuscript has been modified accordingly.

Conerning the different font for Dactylfric acid and Eriodictyol, we have corrected this mistake.

-Chapter 4.3

line 463 enter the duration of the extraction. This parameter was not given in the additional materials either.

# Following your suggestion, this parameter is now specified in the Supplemental File S1.

# Additionally, we have carried out a deep edition and correction of all the language and typos of the manuscript.

# We acknowledge all your suggestions, sincerely, which have deeply helped to improve the manuscript.